# STACKELBERG GAN: TOWARDS PROVABLE MINIMAX EQUILIBRIUM VIA MULTI-GENERATOR ARCHITECTURES

## ABSTRACT

We study the problem of alleviating the instability issue in the GAN training procedure via new architecture design. The discrepancy between the minimax and maximin objective values could serve as a proxy for the difficulties that the alternating gradient descent encounters in the optimization of GANs. In this work, we give new results on the benefits of multi-generator architecture of GANs. We show that the minimax gap shrinks to $\epsilon$ as the number of generators increases with rate $\widetilde{\mathcal{O}}(1/\epsilon)$. This improves over the best-known result of $\widetilde{\mathcal{O}}(1/\epsilon^2)$. At the core of our techniques is a novel application of Shapley-Folkman lemma to the *generic* minimax problem, where in the literature the technique was only known to work when the objective function is restricted to the Lagrangian function of a constraint optimization problem. Our proposed Stackelberg GAN performs well experimentally in both synthetic and real-world datasets, improving Fréchet Inception Distance by $14.61\%$ over the previous multi-generator GANs on the benchmark datasets.

## 1 INTRODUCTION

Generative Adversarial Nets (GANs) are emerging objects of study in machine learning, computer vision, natural language processing, and many other domains. In machine learning, study of such a framework has led to significant advances in adversarial defenses (Xiao et al., 2018; Samangouei et al., 2018) and machine security (Athalye et al., 2018; Samangouei et al., 2018). In computer vision and natural language processing, GANs have resulted in improved performance over standard generative models for images and texts (Goodfellow et al., 2014), such as variational autoencoder (Kingma & Welling, 2013) and deep Boltzmann machine (Salakhutdinov & Larochelle, 2010). A main technique to achieve this goal is to play a minimax two-player game between generator and discriminator under the design that the generator tries to confuse the discriminator with its generated contents and the discriminator tries to distinguish real images/texts from what the generator creates.

Despite a large amount of variants of GANs, many fundamental questions remain unresolved. One of the long-standing challenges is designing *universal, easy-to-implement* architectures that alleviate the instability issue of GANs training. Ideally, GANs are supposed to solve the minimax optimization problem (Goodfellow et al., 2014), but in practice alternating gradient descent methods do not clearly privilege minimax over maximin or vice versa (page 35, Goodfellow (2016)), which may lead to instability in training if there exists a large discrepancy between the minimax and maximin objective values. The focus of this work is on improving the stability of such minimax game in the training process of GANs.

To alleviate the issues caused by the large minimax gap, our study is motivated by the so-called Stackelberg competition in the domain of game theory. In the Stackelberg leadership model, the players of this game are one *leader* and multiple *followers*, where the leader firm moves first and then the follower firms move sequentially. It is known that the Stackelberg model can be solved to find a *subgame perfect Nash equilibrium*. We apply this idea of Stackelberg leadership model to the architecture design of GANs. That is, we design an improved GAN architecture with multiple generators (followers) which team up to play against the discriminator (leader). We therefore name our model *Stackelberg GAN*. Our theoretical and experimental results establish that: *GANs with multi-generator architecture have smaller minimax gap, and enjoy more stable training performances.*

**Our Contributions.** This paper tackles the problem of instability during the GAN training procedure with both theoretical and experimental results. We study this problem by new architecture design.

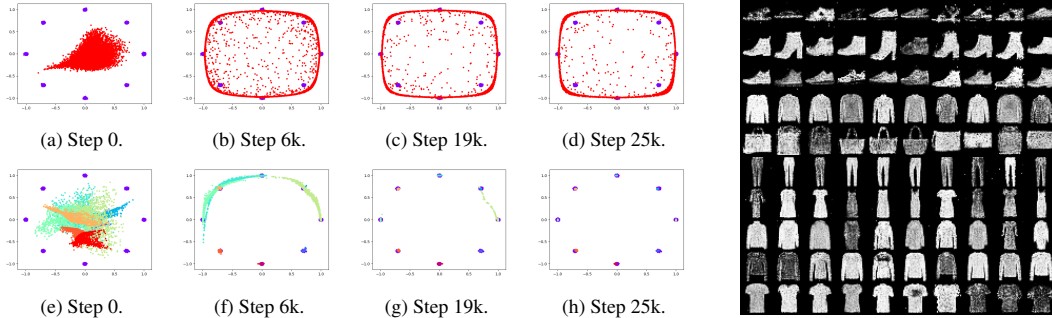

(a) Step 0.     (b) Step 6k.     (c) Step 19k.     (d) Step 25k.

(e) Step 0.     (f) Step 6k.     (g) Step 19k.     (h) Step 25k.

Figure 1: **Left Figure, Top Row:** Standard GAN training on a toy 2D mixture of 8 Gaussians. **Left Figure, Bottom Row:** Stackelberg GAN training with 8 generator ensembles, each of which is denoted by one color. **Right Figure:** Stackelberg GAN training with 10 generator ensembles on fashion-MNIST dataset without cherry pick, where each row corresponds to one generator.

- We propose the Stackelberg GAN framework of multiple generators in the GAN architecture. Our framework is general since it can be applied to all variants of GANs, e.g., vanilla GAN, Wasserstein GAN, etc. It is built upon the idea of *jointly* optimizing an ensemble of GAN losses w.r.t. all pairs of discriminator and generator.

  *Differences from prior work.* Although the idea of having multiple generators in the GAN architecture is not totally new, e.g., MIX+GAN (Arora et al., 2017), MGAN (Hoang et al., 2018), MAD-GAN (Ghosh et al., 2017) and GMAN (Durugkar et al., 2016), there are key differences between Stackelberg GAN and prior work. a) In MGAN (Hoang et al., 2018) and MAD-GAN (Ghosh et al., 2017), various generators are combined as a mixture of probabilistic models with assumption that the generators and discriminator have infinite capacity. Also, they require that the generators share common network parameters. In contrast, in the Stackelberg GAN model we allow various sampling schemes beyond the mixture model, e.g., each generator samples a fixed but unequal number of data points independently. Furthermore, each generator has free parameters. We also make no assumption on the model capacity in our analysis. This is an important research question as raised by Arora et al. (2018). b) In MIX+GAN (Arora et al., 2017), the losses are ensembled with learned weights and an extra regularization term, which discourages the weights being too far away from uniform. We find it slightly unnecessary because the expressive power of each generator already allows *implicit scaling* of each generator. In the Stackelberg GAN, we apply equal weights for all generators and obtain improved guarantees. c) In GMAN (Durugkar et al., 2016), there are multiple discriminators while it is unclear in theory why multi-discriminator architecture works well. In this paper, we provide formal guarantees for our model.

- We prove that the minimax duality gap shrinks as the number of generators increases (see Theorem 1 and Corollary 2). Unlike the previous work, our result has no assumption on the expressive power of generators and discriminator, but instead depends on their non-convexity. With extra condition on the expressive power of generators, we show that Stackelberg GAN is able to achieve $\epsilon$-approximate equilibrium with $\widetilde{\mathcal{O}}(1/\epsilon)$ generators (see Theorem 3). This improves over the best-known result in (Arora et al., 2017) which requires generators as many as $\widetilde{\mathcal{O}}(1/\epsilon^2)$. At the core of our techniques is a novel application of the Shapley-Folkman lemma to the *generic* minimax problem, where in the literature the technique was only known to work when the objective function is restricted to the Lagrangian function of a constrained optimization problem (Zhang et al., 2018). This results in tighter bounds than that of the covering number argument as in (Arora et al., 2017). We also note that MIX+GAN is a heuristic model which does not exactly match the theoretical analysis in (Arora et al., 2017), while this paper provides formal guarantees for the exact model of Stackelberg GAN.

- We empirically study the performance of Stackelberg GAN for various synthetic and real datasets. We observe that without any human assignment, surprisingly, each generator automatically learns balanced number of modes without any mode being dropped (see Figure 1). Compared with other multi-generator GANs with the same network capacity, our experiments show that Stackelberg GAN enjoys 26.76 Fréchet Inception Distance on CIFAR-10 dataset while prior results achieve 31.34 (smaller is better), achieving an improvement of 14.61%.

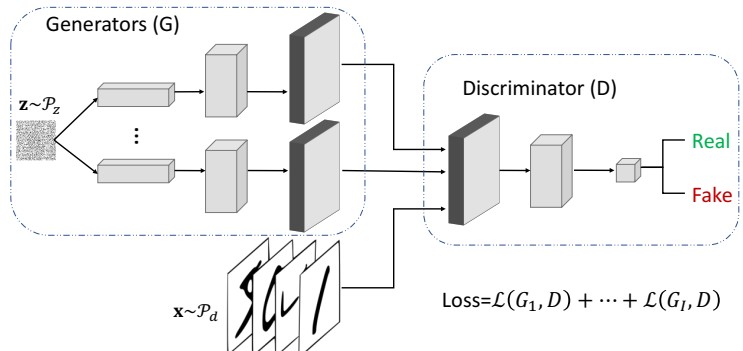

Figure 2: Architecture of Stackelberg GAN. We ensemble the losses of various generator and discriminator pairs with equal weights.

## 2 STACKELBERG GAN

Before proceeding, we define some notations and formalize our model setup in this section.

**Notations.** We will use bold lower-case letter to represent vector and lower-case letter to represent scalar. Specifically, we denote by $\theta \in \mathbb{R}^t$ the parameter vector of discriminator and $\gamma \in \mathbb{R}^g$ the parameter vector of generator. Let $D_\theta(\mathbf{x})$ be the output probability of discriminator given input $\mathbf{x}$, and let $G_\gamma(\mathbf{z})$ represent the generated vector given random input $\mathbf{z}$. For any function $f(\mathbf{u})$, we denote by $f^*(\mathbf{v}) := \sup_{\mathbf{u}}\{\mathbf{u}^T\mathbf{v} - f(\mathbf{u})\}$ the conjugate function of $f$. Let $\check{\mathsf{cl}}f$ be the convex closure of $f$, which is defined as the function whose epigraph is the convex closed hull of that of function $f$. We define $\widehat{\mathsf{cl}}f := -\check{\mathsf{cl}}(-f)$. We will use $I$ to represent the number of generators.

### 2.1 MODEL SETUP

**Preliminaries.** The key ingredient in the standard GAN is to play a *zero-sum two-player* game between a discriminator and a generator — which are often parametrized by deep neural networks in practice — such that the goal of the generator is to map random noise $\mathbf{z}$ to some plausible images/texts $G_\gamma(\mathbf{z})$ and the discriminator $D_\theta(\cdot)$ aims at distinguishing the real images/texts from what the generator creates.

For every parameter implementations $\gamma$ and $\theta$ of generator and discriminator, respectively, denote by the payoff value

$$\phi(\gamma;\theta) := \mathbb{E}_{\mathbf{x}\sim\mathcal{P}_d}f(D_\theta(\mathbf{x})) + \mathbb{E}_{\mathbf{z}\sim\mathcal{P}_z}f(1 - D_\theta(G_\gamma(\mathbf{z}))),$$

where $f(\cdot)$ is some concave, increasing function. Hereby, $\mathcal{P}_d$ is the distribution of true images/texts and $\mathcal{P}_z$ is a noise distribution such as Gaussian or uniform distribution. The standard GAN thus solves the following saddle point problems:

$$\inf_{\gamma\in\mathbb{R}^g}\sup_{\theta\in\mathbb{R}^t}\phi(\gamma;\theta), \qquad \text{or} \qquad \sup_{\theta\in\mathbb{R}^t}\inf_{\gamma\in\mathbb{R}^g}\phi(\gamma;\theta). \tag{1}$$

For different choices of function $f$, problem (1) leads to various variants of GAN. For example, when $f(t) = \log t$, problem (1) is the classic GAN; when $f(t) = t$, it reduces to the Wasserstein GAN. We refer interested readers to the paper of Nowozin et al. (2016) for more variants of GANs.

**Stackelberg GAN.** Our model of Stackelberg GAN is inspired from the Stackelberg competition in the domain of game theory. Instead of playing a two-player game as in the standard GAN, in Stackelberg GAN there are $I + 1$ players with two firms — one discriminator and $I$ generators. One can make an analogy between the discriminator (generators) in the Stackelberg GAN and the leader (followers) in the Stackelberg competition.

Stackelberg GAN is a general framework which can be built on top of all variants of standard GANs. The objective function is simply an ensemble of losses w.r.t. all possible pairs of generators and discriminator: $\Phi(\gamma_1, ..., \gamma_I; \theta) := \sum_{i=1}^{I}\phi(\gamma_i; \theta)$. Thus it is very easy to implement. The Stackelberg GAN therefore solves the following saddle point problems:

$$w^* := \inf_{\gamma_1,...,\gamma_I\in\mathbb{R}^g}\sup_{\theta\in\mathbb{R}^t}\frac{1}{I}\Phi(\gamma_1, ..., \gamma_I; \theta), \qquad \text{or} \qquad q^* := \sup_{\theta\in\mathbb{R}^t}\inf_{\gamma_1,...,\gamma_I\in\mathbb{R}^g}\frac{1}{I}\Phi(\gamma_1, ..., \gamma_I; \theta).$$

We term $w^* - q^*$ the *minimax (duality) gap*. We note that there are key differences between the naïve ensembling model and ours. In the naïve ensembling model, one trains multiple GAN models *independently* and averages their outputs. In contrast, our Stackelberg GAN shares a unique discriminator for various generators, thus requires *jointly training*. Figure 2 shows the architecture of our Stackelberg GAN.

**How to generate samples from Stackelberg GAN?** In the Stackelberg GAN, we expect that each generator learns only a few modes. In order to generate a sample that may come from all modes, we use a mixed model. In particular, we generate a uniformly random value $i$ from 1 to $I$ and use the $i$-th generator to obtain a new sample. Note that this procedure in independent of the training procedure.

## 3 ANALYSIS OF STACKELBERG GAN

In this section, we develop our theoretical contributions and compare our results with the prior work.

### 3.1 MINIMAX DUALITY GAP

We begin with studying the minimax gap of Stackelberg GAN. Our main results show that the minimax gap shrinks as the number of generators increases.

To proceed, denote by $h_i(\mathbf{u}_i) := \inf_{\gamma_i \in \mathbb{R}^g} (-\phi(\gamma_i; \cdot))^*(\mathbf{u}_i)$, where the conjugate operation is w.r.t. the second argument of $\phi(\gamma_i; \cdot)$. We clarify here that the subscript $i$ in $h_i$ indicates that the function $h_i$ is derived from the $i$-th generator. The argument of $h_i$ should depend on $i$, so we denote it by $\mathbf{u}_i$. Intuitively, $h_i$ serves as an approximate convexification of $-\phi(\gamma_i, \cdot)$ w.r.t the second argument due to the conjugate operation. Denote by $\breve{\mathsf{cl}}h_i$ the convex closure of $h_i$:

$$\breve{\mathsf{cl}}h_i(\widetilde{\mathbf{u}}) := \inf_{\{a^j\}, \{\mathbf{u}_i^j\}} \left\{ \sum_{j=1}^{t+2} a^j h_i(\mathbf{u}_i^j) : \widetilde{\mathbf{u}} = \sum_{j=1}^{t+2} a^j \mathbf{u}_i^j, \sum_{j=1}^{t+2} a^j = 1, a^j \geq 0 \right\}.$$

$\breve{\mathsf{cl}}h_i$ represents the convex relaxation of $h_i$ because the epigraph of $\breve{\mathsf{cl}}h_i$ is exactly the convex hull of epigraph of $h_i$ by the definition of $\breve{\mathsf{cl}}h_i$. Let $\Delta_\theta^{\mathsf{minimax}} = \inf_{\gamma_1,...,\gamma_I \in \mathbb{R}^g} \sup_{\theta \in \mathbb{R}^t} \frac{1}{I}\Phi(\gamma_1,...,\gamma_I; \theta) - \inf_{\gamma_1,...,\gamma_I \in \mathbb{R}^g} \sup_{\theta \in \mathbb{R}^t} \frac{1}{I}\widetilde{\Phi}(\gamma_1,...,\gamma_I; \theta)$, and $\Delta_\theta^{\mathsf{maximin}} = \sup_{\theta \in \mathbb{R}^t} \inf_{\gamma_1,...,\gamma_I \in \mathbb{R}^g} \frac{1}{I}\widetilde{\Phi}(\gamma_1,...,\gamma_I; \theta) - \sup_{\theta \in \mathbb{R}^t} \inf_{\gamma_1,...,\gamma_I \in \mathbb{R}^g} \frac{1}{I}\Phi(\gamma_1,...,\gamma_I; \theta)$, where $\widetilde{\Phi}(\gamma_1,...,\gamma_I; \theta) := \sum_{i=1}^I \widehat{\mathsf{cl}}\phi(\gamma_i; \theta)$ and $-\widehat{\mathsf{cl}}\phi(\gamma_i; \theta)$ is the convex closure of $-\phi(\gamma_i; \theta)$ w.r.t. argument $\theta$. Therefore, $\Delta_\theta^{\mathsf{maximin}} + \Delta_\theta^{\mathsf{minimax}}$ measures the non-convexity of objective function w.r.t. argument $\theta$. For example, it is equal to 0 if and only if $\phi(\gamma_i; \theta)$ is concave and closed w.r.t. discriminator parameter $\theta$.

We have the following guarantees on the minimax gap of Stackelberg GAN.

**Theorem 1.** *Let $\Delta_\gamma^i := \sup_{\mathbf{u} \in \mathbb{R}^t} \{h_i(\mathbf{u}) - \breve{\mathsf{cl}}h_i(\mathbf{u})\} \geq 0$ and $\Delta_\gamma^{\mathsf{worst}} := \max_{i \in [I]} \Delta_\gamma^i$. Denote by $t$ the number of parameters of discriminator, i.e., $\theta \in \mathbb{R}^t$. Suppose that $h_i(\cdot)$ is continuous and $\mathsf{dom}h_i$ is compact and convex. Then the duality gap can be bounded by*

$$0 \leq w^* - q^* \leq \Delta_\theta^{\mathsf{minimax}} + \Delta_\theta^{\mathsf{maximin}} + \epsilon,$$

*provided that the number of generators $I > \frac{t+1}{\epsilon} \Delta_\gamma^{\mathsf{worst}}$.*

**Remark 1.** *Theorem 1 makes mild assumption on the continuity of loss and no assumption on the model capacity of discriminator and generators. The analysis instead depends on their non-convexity as being parametrized by deep neural networks. In particular, $\Delta_\gamma^i$ measures the divergence between the function value of $h_i$ and its convex relaxation $\breve{\mathsf{cl}}h_i$; When $\phi(\gamma_i; \theta)$ is convex w.r.t. argument $\gamma_i$, $\Delta_\gamma^i$ is exactly 0. The constant $\Delta_\gamma^{\mathsf{worst}}$ is the maximal divergence among all generators, which does not grow with the increase of $I$. This is because $\Delta_\gamma^{\mathsf{worst}}$ measures the divergence of only* one *generator and when each generator for example has the same architecture, we have $\Delta_\gamma^{\mathsf{worst}} = \Delta_\gamma^1 = ... = \Delta_\gamma^I$. Similarly, the terms $\Delta_\theta^{\mathsf{minimax}}$ and $\Delta_\theta^{\mathsf{maximin}}$ characterize the non-convexity of discriminator. When the discriminator is concave such as logistic regression and support vector machine, $\Delta_\theta^{\mathsf{minimax}} = \Delta_\theta^{\mathsf{maximin}} = 0$ and we have the following straightforward corollary about the minimax duality gap of Stackelberg GAN.*

**Corollary 2.** *Under the settings of Theorem 1, when $\phi(\gamma_i; \theta)$ is concave and closed w.r.t. discriminator parameter $\theta$ and the number of generators $I > \frac{t+1}{\epsilon} \Delta_\gamma^{\mathsf{worst}}$, we have $0 \leq w^* - q^* \leq \epsilon$.*

## 3.2 EXISTENCE OF APPROXIMATE EQUILIBRIUM

The results of Theorem 1 and Corollary 2 are independent of model capacity of generators and discriminator. When we make assumptions on the expressive power of generator as in (Arora et al., 2017), we have the following guarantee (2) on the existence of $\epsilon$-approximate equilibrium.

**Theorem 3.** *Under the settings of Theorem 1, suppose that for any $\xi > 0$, there exists a generator $G$ such that $\mathbb{E}_{\mathbf{x} \sim \mathcal{P}_d, \mathbf{z} \sim \mathcal{P}_{\mathbf{z}}} \|G(\mathbf{z}) - \mathbf{x}\|_2 \leq \xi$. Let the discriminator and the generators be $L$-Lipschitz w.r.t. inputs and parameters, respectively. Then for any $\epsilon > 0$, there exist $I = \frac{t+1}{\epsilon} \Delta_\gamma^{\text{worst}}$ generators $G_{\gamma_1^*}, ..., G_{\gamma_I^*}$ and a discriminator $D_{\theta^*}$ such that for some value $V \in \mathbb{R}$,*

$$
\begin{aligned}
\forall \gamma_1, ..., \gamma_I \in \mathbb{R}^g, \quad & \Phi(\gamma_1, ..., \gamma_I; \theta^*) \leq V + \epsilon, \\
\forall \theta \in \mathbb{R}^t, \quad & \Phi(\gamma_1^*, ..., \gamma_I^*; \theta) \geq V - \epsilon.
\end{aligned}
\tag{2}
$$

**Related Work.** While many efforts have been devoted to empirically investigating the performance of multi-generator GAN, little is known about how many generators are needed so as to achieve certain equilibrium guarantees. Probably the most relevant prior work to Theorem 3 is that of (Arora et al., 2017). In particular, Arora et al. (2017) showed that there exist $I = \frac{100t}{\epsilon^2} \Delta^2$ generators and one discriminator such that $\epsilon$-approximate equilibrium can be achieved, provided that for *all* $\mathbf{x}$ and any $\xi > 0$, there exists a generator $G$ such that $\mathbb{E}_{\mathbf{z} \sim \mathcal{P}_{\mathbf{z}}} \|G(\mathbf{z}) - \mathbf{x}\|_2 \leq \xi$. Hereby, $\Delta$ is a global upper bound of function $|f|$, i.e., $f \in [-\Delta, \Delta]$. In comparison, Theorem 3 improves over this result in two aspects: a) the assumption on the expressive power of generators in (Arora et al., 2017) implies our condition $\mathbb{E}_{\mathbf{x} \sim \mathcal{P}_d, \mathbf{z} \sim \mathcal{P}_{\mathbf{z}}} \|G(\mathbf{z}) - \mathbf{x}\|_2 \leq \xi$. Thus our assumption is weaker. b) The required number of generators in Theorem 3 is as small as $\frac{t+1}{\epsilon} \Delta_\gamma^{\text{worst}}$. We note that $\Delta_\gamma^{\text{worst}} \ll 2\Delta$ by the definition of $\Delta_\gamma^{\text{worst}}$. Therefore, Theorem 3 requires much fewer generators than that of (Arora et al., 2017).

## 4 ARCHITECTURE, CAPACITY AND MODE COLLAPSE/DROPPING

In this section, we empirically investigate the effect of network architecture and capacity on the mode collapse/dropping issues for various multi-generator architecture designs. Hereby, the *mode dropping* refers to the phenomenon that generative models simply ignore some hard-to-represent modes of real distributions, and the *mode collapse* means that some modes of real distributions are "averaged" by generative models. For GAN, it is widely believed that the two issues are caused by the large gap between the minimax and maximin objective function values (see page 35, Goodfellow (2016)).

Our experiments verify that network capacity (change of width and depth) is not very crucial for resolving the mode collapse issue, though it can alleviate the mode dropping in certain senses. Instead, the choice of architecture of generators plays a key role. To visualize this discovery, we test the performance of varying architectures of GANs on a synthetic mixture of Gaussians dataset with 8 modes and 0.01 standard deviation. We observe the following phenomena:

**Naïvely increasing capacity of one-generator architecture does not alleviate mode collapse.** It shows that the multi-generator architecture in the Stackelberg GAN effectively alleviates the mode collapse issue. Though naïvely increasing capacity of one-generator architecture alleviates mode dropping issue, for more challenging mode collapse issue, the effect is not obvious (see Figure 3).

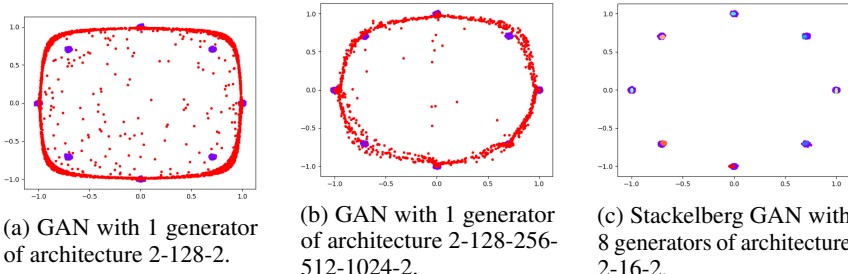

(a) GAN with 1 generator of architecture 2-128-2.

(b) GAN with 1 generator of architecture 2-128-256-512-1024-2.

(c) Stackelberg GAN with 8 generators of architecture 2-16-2.

Figure 3: Comparison of mode collapse/dropping issue of one-generator and multi-generator architectures with varying model capacities. (a) and (b) show that increasing the model capacity can alleviate the mode dropping issue, though it does not alleviate the mode collapse issue. (c) Multi-generator architecture with even small capacity resolves the mode collapse issue.

**Stackelberg GAN outperforms multi-branch models.** We compare performance of multi-branch GAN and Stackelberg GAN with objective functions:

$$
\text{(Multi-Branch GAN)} \quad \phi\left(\frac{1}{I}\sum_{i=1}^{I} \gamma_i; \theta\right) \qquad \text{vs.} \qquad \text{(Stackelberg GAN)} \quad \frac{1}{I}\sum_{i=1}^{I} \phi(\gamma_i; \theta).
$$

Hereby, the multi-branch GAN has made use of extra information that the real distribution is Gaussian mixture model with probability distribution function $\frac{1}{I} \sum_{i=1}^{I} p_{\mathcal{N}_i}(\mathbf{x})$, so that each $\gamma_i$ tries to fit one component. However, even this we observe that with same model capacity, Stackelberg GAN significantly outperforms multi-branch GAN (see Figure 4 (a)(c)) even without access to the extra information. The performance of Stackelberg GAN is also better than multi-branch GAN of much larger capacity (see Figure 4 (b)(c)).

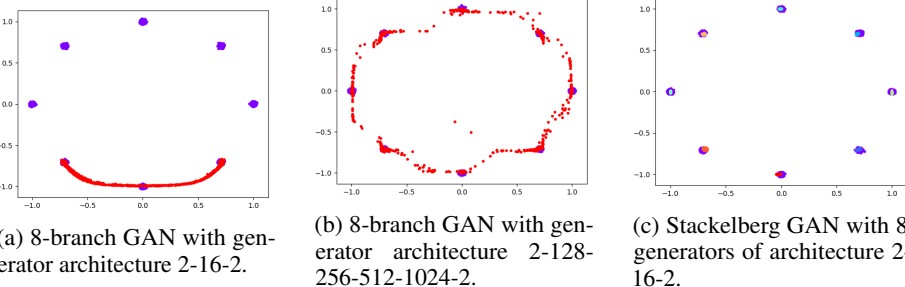

(a) 8-branch GAN with generator architecture 2-16-2.

(b) 8-branch GAN with generator architecture 2-128-256-512-1024-2.

(c) Stackelberg GAN with 8 generators of architecture 2-16-2.

Figure 4: Comparison of mode collapse issue of multi-branch and multi-generator architectures with varying model capacities. (a) and (b) show that increasing the model capacity can alleviate the mode dropping issue, though it does not alleviate the mode collapse issue. (c) Multi-generator architecture with much smaller capacity resolves the mode collapse issue.

**Generators tend to learn balanced number of modes when they have same capacity.** We observe that for varying number of generators, each generator in the Stackelberg GAN tends to learn equal number of modes when the modes are symmetric and every generator has same capacity (see Figure 5).

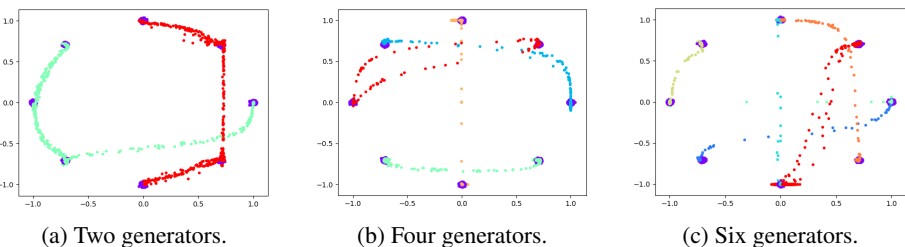

(a) Two generators.

(b) Four generators.

(c) Six generators.

Figure 5: Stackelberg GAN with varying number of generators of architecture 2-128-256-512-1024-2.

## 5 EXPERIMENTS

In this section, we verify our theoretical contributions by the experimental validation.

### 5.1 MNIST DATASET

We first show that Stackelberg GAN generates more diverse images on the MNIST dataset (LeCun et al., 1998) than classic GAN. We follow the standard preprocessing step that each pixel is normalized via subtracting it by 0.5 and dividing it by 0.5. The detailed network setups of discriminator and generators are in Table 4.

Figure 6 shows the diversity of generated digits by Stackelberg GAN with varying number of generators. When there is only one generator, the digits are not very diverse with many repeated "1"'s and much fewer "2"'s. As the number of generators increases, the generated images tend to be more diverse. In particular, for 10-generator Stackelberg GAN, each generator is associated with one or two digits without any digit being missed.

### 5.2 FASHION-MNIST DATASET

We also observe better performance by the Stackelberg GAN on the Fashion-MNIST dataset. Fashion-MNIST is a dataset which consists of 60,000 examples. Each example is a $28 \times 28$ grayscale image associating with a label from 10 classes. We follow the standard preprocessing step that each pixel is normalized via subtracting it by 0.5 and dividing it by 0.5. We specify the detailed network setups of discriminator and generators in Table 4.

Figure 7 shows the diversity of generated fashions by Stackelberg GAN with varying number of generators. When there is only one generator, the generated images are not very diverse without

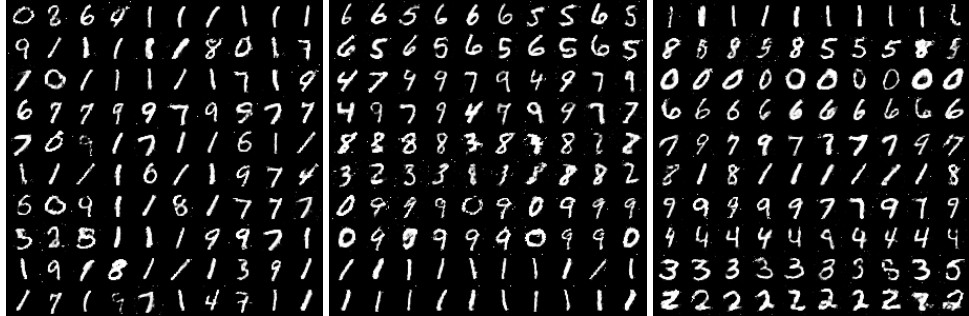

Figure 6: Standard GAN vs. Stackelberg GAN on the MNIST dataset without cherry pick. **Left Figure:** Digits generated by the standard GAN. It shows that the standard GAN generates many "1"'s which are not very diverse. **Middle Figure:** Digits generated by the Stackelberg GAN with 5 generators, where every two rows correspond to one generator. **Right Figure:** Digits generated by the Stackelberg GAN with 10 generators, where each row corresponds to one generator.

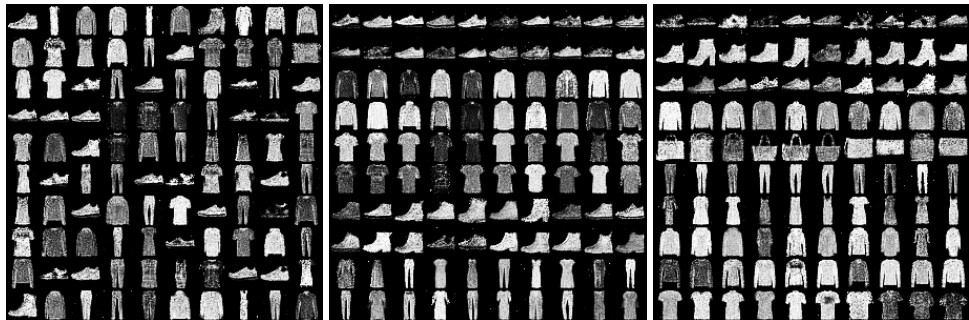

Figure 7: Generated samples by Stackelberg GAN on CIFAR-10 dataset without cherry pick. **Left Figure:** Examples generated by the standard GAN. It shows that the standard GAN fails to generate bags. **Middle Figure:** Examples generated by the Stackelberg GAN with 5 generators, where every two rows correspond to one generator. **Right Figure:** Examples generated by the Stackelberg GAN with 10 generators, where each row corresponds to one generator.

any bags being found. As the number of generators increases, the generated images tend to be more diverse. In particular, for 10-generator Stackelberg GAN, each generator is associated with one class without any class being missed.

## 5.3 CIFAR-10 DATASET

We then implement Stackelberg GAN on the CIFAR-10 dataset. CIFAR-10 includes 60,000 32×32 training images, which fall into 10 classes (Krizhevsky & Hinton, 2009)). The architecture of generators and discriminator follows the design of DCGAN in (Radford et al., 2015). We train models with 5, 10, and 20 fixed-size generators. The results show that the model with 10 generators performs the best. We also train 10-generator models where each generator has 2, 3 and 4 convolution layers. We find that the generator with 2 convolution layers, which is the most shallow one, performs the best. So we report the results obtained from the model with 10 generators containing 2 convolution layers. Figure 8a shows the samples produced by different generators. The samples are randomly drawn instead of being cherry-picked to demonstrate the quality of images generated by our model.

For quantitative evaluation, we use Inception score and Fréchet Inception Distance (FID) to measure the difference between images generated by models and real images.

**Results of Inception Score.** The Inception score measures the quality of a generated image and is correlated well with human's judgment (Salimans et al., 2016). We report the Inception score obtained by our Stackelberg GAN and other baseline methods in Table 1. For fair comparison, we only consider the baseline models which are completely unsupervised model and do not need any label information. Instead of directly using the reported Inception scores by original papers, we replicate the experiment of *MGAN* using the code, architectures and parameters reported by their original papers, and evaluate the scores based on the new experimental results. Table 1 shows that our model achieves a score of 7.62 in CIFAR-10 dataset, which outperforms the state-of-the-art models.

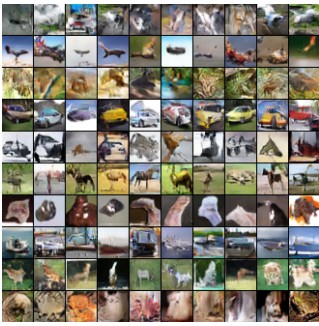 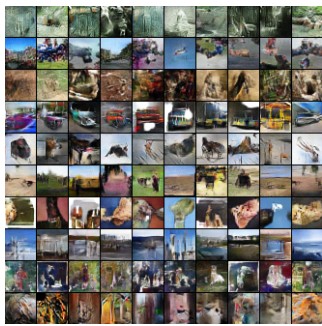

(a) Samples on CIFAR-10.  (b) Samples on Tiny ImageNet.

Figure 8: Examples generated by Stackelberg GAN on CIFAR-10 (left) and Tiny ImageNet (right) without cherry pick, where each row corresponds to samples from one generator.

For fairness, we configure our Stackelberg GAN with the same capacity as MGAN, that is, the two models have comparative number of total parameters. When the capacity of our Stackelberg GAN is as small as DCGAN, our model improves over DCGAN significantly.

**Results of Fréchet Inception Distance.** We then evaluate the performance of models on CIFAR-10 dataset using the Fréchet Inception Distance (FID), which better captures the similarity between generated images and real ones (Heusel et al., 2017). As Table 1 shows, under the same capacity as DCGAN, our model reduces the FID by $20.74\%$. Meanwhile, under the same capacity as MGAN, our model reduces the FID by $14.61\%$. This improvement further indicates that our Stackelberg GAN with multiple light-weight generators help improve the quality of the generated images.

Table 1: Quantitative evaluation of various GANs on CIFAR-10 dataset. All results are either reported by the authors themselves or run by us with codes provided by the authors. Every model is trained *without label*. Methods with higher inception score and lower Fréchet Inception Distance are better.

| Model | Inception Score | Fréchet Inception Distance |
|---|:---:|:---:|
| Real data | $11.24 \pm 0.16$ | - |
| WGAN (Arjovsky et al., 2017) | $3.82 \pm 0.06$ | - |
| MIX+WGAN (Arora et al., 2017) | $4.04 \pm 0.07$ | - |
| Improved-GAN (Salimans et al., 2016) | $4.36 \pm 0.04$ | - |
| ALI (Dumoulin et al., 2017) | $5.34 \pm 0.05$ | - |
| BEGAN (Berthelot et al., 2017) | $5.62$ | - |
| MAGAN (Wang et al., 2017) | $5.67$ | - |
| GMAN (Durugkar et al., 2016) | $6.00 \pm 0.19$ | - |
| DCGAN (Radford et al., 2015) | $6.40 \pm 0.05$ | 37.7 |
| **Ours (capacity as DCGAN)** | $\mathbf{7.02 \pm 0.07}$ | **29.88** |
| D2GAN (Nguyen et al., 2017) | $7.15 \pm 0.07$ | - |
| MAD-GAN (our run) (Ghosh et al., 2017) | $6.67 \pm 0.07$ | 34.10 |
| MGAN (our run) (Hoang et al., 2018) | $7.52 \pm 0.1$ | 31.34 |
| **Ours (capacity $1\times$MGAN$\approx 1.8\times$DCGAN)** | $\mathbf{7.62 \pm 0.07}$ | **26.76** |

### 5.4 TINY IMAGENET DATASET

We also evaluate the performance of Stackelberg GAN on the Tiny ImageNet dataset. The Tiny ImageNet is a large image dataset, where each image is labelled to indicate the class of the object inside the image. We resize the figures down to $32 \times 32$ following the procedure described in (Chrabaszcz et al., 2017). Figure 8b shows the randomly picked samples generated by 10-generator Stackelberg GAN. Each row has samples generated from one generator. Since the types of some images in the Tiny ImageNet are also included in the CIFAR-10, we order the rows in the similar way as Figure 8a .

## 6 CONCLUSIONS

In this work, we tackle the problem of instability during GAN training procedure, which is caused by the huge gap between minimax and maximin objective values. The core of our techniques is a multi-generator architecture. We show that the minimax gap shrinks to $\epsilon$ as the number of generators increases with rate $\widetilde{\mathcal{O}}(1/\epsilon)$, when the maximization problem w.r.t. the discriminator is concave. This improves over the best-known results of $\widetilde{\mathcal{O}}(1/\epsilon^2)$. Experiments verify the effectiveness of our proposed methods.

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

# A  SUPPLEMENTARY EXPERIMENTS

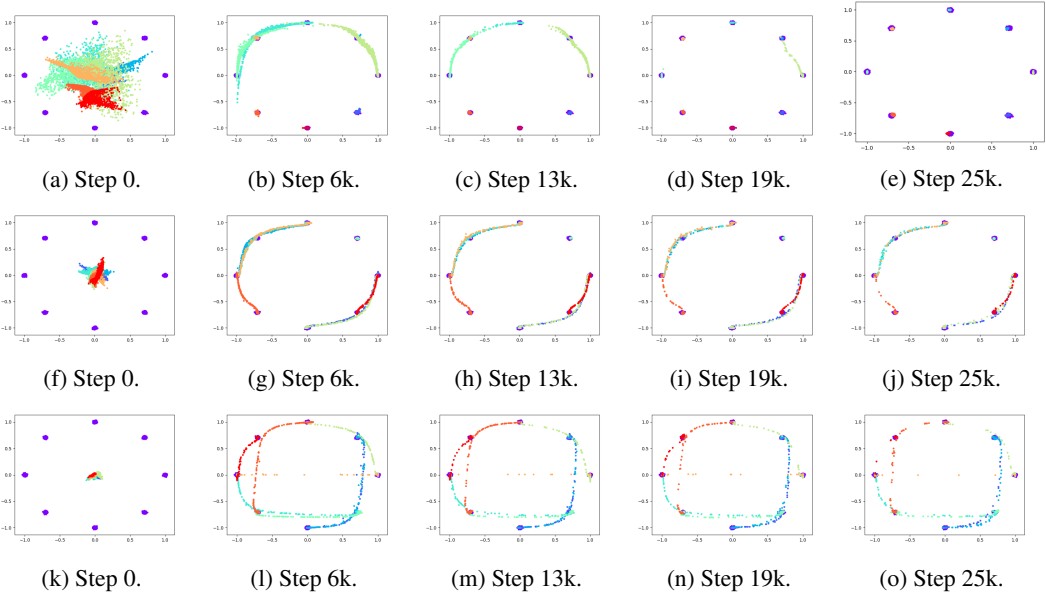

(a) Step 0.  (b) Step 6k.  (c) Step 13k.  (d) Step 19k.  (e) Step 25k.

(f) Step 0.  (g) Step 6k.  (h) Step 13k.  (i) Step 19k.  (j) Step 25k.

(k) Step 0.  (l) Step 6k.  (m) Step 13k.  (n) Step 19k.  (o) Step 25k.

Figure 9: Effects of generator architecture of Stackelberg GAN on a toy 2D mixture of Gaussians, where the number of generators is set to be 8. **Top Row:** The generators have one hidden layer. **Middle Row:** The generators have two hidden layers. **Bottom Row:** The generators have three hidden layer. It shows that with the number of hidden layers increasing, each generator tends to learn more modes. However, mode collapse never happens for all three architectures.

Figure 9 shows how the architecture of generators affects the distributions of samples by each generators. The enlarged versions of samples generated by Stackelberg GAN with architectures shown in Table 5 and Table 6 are deferred to Figures 10, 11, 12 and 13.

# B  PROOFS OF MAIN RESULTS

## B.1  PROOFS OF THEOREM 1 AND COROLLARY 2: MINIMAX DUALITY GAP

The statement $0 \leq w^* - q^*$ is by the weak duality. Thus it suffices to prove the other side of the inequality. All notations in this section are defined in Section 3.1.

We first show that

$$\inf_{\gamma_1,...,\gamma_I \in \mathbb{R}^g} \sup_{\theta \in \mathbb{R}^t} \frac{1}{I} \widetilde{\Phi}(\gamma_1,...,\gamma_I; \theta) - \sup_{\theta \in \mathbb{R}^t} \inf_{\gamma_1,...,\gamma_I \in \mathbb{R}^g} \frac{1}{I} \widetilde{\Phi}(\gamma_1,...,\gamma_I; \theta) \leq \epsilon.$$

Denote by

$$p(\mathbf{u}) := \inf_{\gamma_1,...,\gamma_I \in \mathbb{R}^g} \sup_{\theta \in \mathbb{R}^t} \left\{ \widetilde{\Phi}(\gamma_1,...,\gamma_I; \theta) - \mathbf{u}^T \theta \right\}.$$

We have the following lemma.

**Lemma 4.** *We have*

$$\sup_{\theta \in \mathbb{R}^t} \inf_{\gamma_1,...,\gamma_I \in \mathbb{R}^g} \widetilde{\Phi}(\gamma_1,...,\gamma_I; \theta) = (\check{\mathsf{c}} \mathsf{l} p)(\mathbf{0}) \leq p(\mathbf{0}) = \inf_{\gamma_1,...,\gamma_I \in \mathbb{R}^g} \sup_{\theta \in \mathbb{R}^t} \widetilde{\Phi}(\gamma_1,...,\gamma_I; \theta).$$

*Proof.* By the definition of $p(\mathbf{0})$, we have $p(\mathbf{0}) = \inf_{\gamma_1,...,\gamma_I \in \mathbb{R}^g} \sup_{\theta \in \mathbb{R}^t} \widetilde{\Phi}(\gamma_1,...,\gamma_I; \theta)$. Since $(\check{\mathsf{c}} \mathsf{l} p)(\cdot)$ is the convex closure of function $p(\cdot)$ (a.k.a. weak duality theorem), we have $(\check{\mathsf{c}} \mathsf{l} p)(\mathbf{0}) \leq p(\mathbf{0})$. We now show that

$$\sup_{\theta \in \mathbb{R}^t} \inf_{\gamma_1,...,\gamma_I \in \mathbb{R}^g} \widetilde{\Phi}(\gamma_1,...,\gamma_I; \theta) = (\check{\mathsf{c}} \mathsf{l} p)(\mathbf{0}).$$

Note that $p(\mathbf{u}) = \inf_{\gamma_1,...,\gamma_I \in \mathbb{R}^g} p_{\gamma_1,...,\gamma_I}(\mathbf{u})$, where $p_{\gamma_1,...,\gamma_I}(\mathbf{u}) = \sup_{\theta \in \mathbb{R}^t} \{\widetilde{\Phi}(\gamma_1, ..., \gamma_I; \theta) - \mathbf{u}^T\theta\} = (-\widetilde{\Phi}(\gamma_1, ..., \gamma_I; \cdot))^*(-\mathbf{u})$, and that

$$
\begin{aligned}
\inf_{\mathbf{u} \in \mathbb{R}^t} \{p_{\gamma_1,...,\gamma_I}(\mathbf{u}) + \mathbf{u}^T\mu\} &= -\sup_{\mathbf{u} \in \mathbb{R}^t} \{\mathbf{u}^T(-\mu) - p_{\gamma_1,...,\gamma_I}(\mathbf{u})\} \\
&= -(p_{\gamma_1,...,\gamma_I})^*(-\mu) \quad \text{(by the definition of conjugate function)} \\
&= -(-\widetilde{\Phi}(\gamma_1, ..., \gamma_I; \cdot))^{**}(\mu) \\
&= \widetilde{\Phi}(\gamma_1, ..., \gamma_I; \mu). \quad \text{(by conjugate theorem)}
\end{aligned}
\tag{3}
$$

So we have

$$
\begin{aligned}
(\breve{\mathsf{cl}}p)(\mathbf{0}) &= \sup_{\mu \in \mathbb{R}^t} \inf_{\mathbf{u} \in \mathbb{R}^t} \{p(\mathbf{u}) + \mathbf{u}^T\mu\} \quad \text{(by Lemma 8)} \\
&= \sup_{\mu \in \mathbb{R}^t} \inf_{\mathbf{u} \in \mathbb{R}^t} \inf_{\gamma_1,...,\gamma_I \in \mathbb{R}^g} \{p_{\gamma_1,...,\gamma_I}(\mathbf{u}) + \mathbf{u}^T\mu\} \quad \text{(by the definition of } p(\mathbf{u})) \\
&= \sup_{\mu \in \mathbb{R}^t} \inf_{\gamma_1,...,\gamma_I \in \mathbb{R}^g} \inf_{\mathbf{u} \in \mathbb{R}^t} \{p_{\gamma_1,...,\gamma_I}(\mathbf{u}) + \mathbf{u}^T\mu\} \\
&= \sup_{\mu \in \mathbb{R}^t} \inf_{\gamma_1,...,\gamma_I \in \mathbb{R}^g} \widetilde{\Phi}(\gamma_1, ..., \gamma_I; \mu). \quad \text{(by Eqn. (3))}
\end{aligned}
$$

$\square$

By Lemma 4, it suffices to show $p(\mathbf{0}) - (\breve{\mathsf{cl}}p)(\mathbf{0}) \le (t+1)\Delta_\gamma^{\mathsf{worst}}$. We have the following lemma.

**Lemma 5.** *Under the assumption in Theorem 1, $p(\mathbf{0}) - (\breve{\mathsf{cl}}p)(\mathbf{0}) \le (t+1)\Delta_\gamma^{\mathsf{worst}}$.*

*Proof.* We note that

$$
\begin{aligned}
p(\mathbf{u}) &:= \inf_{\gamma_1,...,\gamma_I \in \mathbb{R}^g} \sup_{\theta \in \mathbb{R}^t} \left\{ \widetilde{\Phi}(\gamma_1, ..., \gamma_I; \theta) - \mathbf{u}^T\theta \right\} \\
&= \inf_{\gamma_1,...,\gamma_I \in \mathbb{R}^g} \sup_{\theta \in \mathbb{R}^t} \left\{ \sum_{i=1}^I \widehat{\mathsf{cl}}\phi(\gamma_i; \theta) - \mathbf{u}^T\theta \right\} \quad \text{(by the definition of } \widetilde{\Phi}) \\
&= \inf_{\gamma_1,...,\gamma_I \in \mathbb{R}^g} \left( \sum_{i=1}^I -\widehat{\mathsf{cl}}\phi(\gamma_i; \cdot) \right)^* (-\mathbf{u}) \quad \text{(by the definition of conjugate function)} \\
&= \inf_{\gamma_1,...,\gamma_I \in \mathbb{R}^g} \inf_{\mathbf{u}_1+...+\mathbf{u}_I=-\mathbf{u}} \left\{ \sum_{i=1}^I (-\widehat{\mathsf{cl}}\phi(\gamma_i; \cdot))^*(\mathbf{u}_i) \right\} \quad \text{(by Lemma 7)} \\
&= \inf_{\gamma_1,...,\gamma_I \in \mathbb{R}^g} \inf_{\mathbf{u}_1+...+\mathbf{u}_I=-\mathbf{u}} \left\{ \sum_{i=1}^I (-\phi(\gamma_i; \cdot))^*(\mathbf{u}_i) \right\} \quad \text{(by conjugate theorem)} \\
&= \inf_{\mathbf{u}_1+...+\mathbf{u}_I=-\mathbf{u}} \inf_{\gamma_1,...,\gamma_I \in \mathbb{R}^g} \left\{ (-\phi(\gamma_1; \cdot))^*(\mathbf{u}_1) + ... + (-\phi(\gamma_I; \cdot))^*(\mathbf{u}_I) \right\} \\
&=: \inf_{\mathbf{u}_1+...+\mathbf{u}_I=-\mathbf{u}} \{h_1(\mathbf{u}_1) + ... + h_I(\mathbf{u}_I)\}, \quad \text{(by the definition of } h_i(\cdot))
\end{aligned}
$$

where $\mathbf{u}_1, ..., \mathbf{u}_I, \mathbf{u} \in \mathbb{R}^t$. Therefore,

$$
p(\mathbf{0}) = \inf_{\mathbf{u}_1,...,\mathbf{u}_I \in \mathbb{R}^t} \sum_{i=1}^I h_i(\mathbf{u}_i), \quad \text{s.t.} \quad \sum_{i=1}^I \mathbf{u}_i = \mathbf{0}.
$$

Consider the subset of $\mathbb{R}^{t+1}$:

$$
\mathcal{Y}_i := \left\{ \mathbf{y}_i \in \mathbb{R}^{t+1} : \mathbf{y}_i = [\mathbf{u}_i, h_i(\mathbf{u}_i)], \mathbf{u}_i \in \mathsf{dom}h_i \right\}, \quad i \in [I].
$$

Define the vector summation

$$
\mathcal{Y} := \mathcal{Y}_1 + \mathcal{Y}_2 + ... + \mathcal{Y}_I.
$$

Since $h_i(\cdot)$ is continuous and $\mathsf{dom}h_i$ is compact, the set

$$
\{(\mathbf{u}_i, h_i(\mathbf{u}_i)) : \mathbf{u}_i \in \mathsf{dom}h_i\}
$$

is compact. So $\mathcal{Y}$, conv($\mathcal{Y}$), $\mathcal{Y}_i$, and conv($\mathcal{Y}_i$), $i \in [I]$ are all compact sets. According to the definition of $\mathcal{Y}$ and the standard duality argument (Bertsekas, 2009), we have

$$p(\mathbf{0}) = \inf \{w : \text{there exists } (\mathbf{r}, w) \in \mathcal{Y} \text{ such that } \mathbf{r} = \mathbf{0}\},$$

and

$$\check{\mathsf{cl}}p(\mathbf{0}) = \inf \{w : \text{there exists } (\mathbf{r}, w) \in \text{conv}(\mathcal{Y}) \text{ such that } \mathbf{r} = \mathbf{0}\}.$$

We are going to apply the following Shapley-Folkman lemma.

**Lemma 6** (Shapley-Folkman, Starr (1969)). *Let $\mathcal{Y}_i, i \in [I]$ be a collection of subsets of $\mathbb{R}^m$. Then for every $\mathbf{y} \in \text{conv}(\sum_{i=1}^{I} \mathcal{Y}_i)$, there is a subset $\mathcal{I}(\mathbf{y}) \subseteq [I]$ of size at most $m$ such that*

$$\mathbf{y} \in \left[ \sum_{i \notin \mathcal{I}(\mathbf{y})} \mathcal{Y}_i + \sum_{i \in \mathcal{I}(\mathbf{y})} \text{conv}(\mathcal{Y}_i) \right].$$

We apply Lemma 6 to prove Lemma 5 with $m = t + 1$. Let $(\overline{\mathbf{r}}, \overline{w}) \in \text{conv}(\mathcal{Y})$ be such that

$$\overline{\mathbf{r}} = \mathbf{0}, \quad \text{and} \quad \overline{w} = \check{\mathsf{cl}}p(\mathbf{0}).$$

Applying the above Shapley-Folkman lemma to the set $\mathcal{Y} = \sum_{i=1}^{I} \mathcal{Y}_i$, we have that there are a subset $\overline{\mathcal{I}} \subseteq [I]$ of size $t + 1$ and vectors

$$(\overline{\mathbf{r}}_i, \overline{w}_i) \in \text{conv}(\mathcal{Y}_i), \quad i \in \overline{\mathcal{I}} \qquad \text{and} \qquad \overline{\mathbf{u}}_i \in \text{dom}h_i, \quad i \notin \overline{\mathcal{I}},$$

such that

$$\sum_{i \notin \overline{\mathcal{I}}} \overline{\mathbf{u}}_i + \sum_{i \in \overline{\mathcal{I}}} \overline{\mathbf{r}}_i = \overline{\mathbf{r}} = \mathbf{0}, \tag{4}$$

$$\sum_{i \notin \overline{\mathcal{I}}} h_i(\overline{\mathbf{u}}_i) + \sum_{i \in \overline{\mathcal{I}}} \overline{w}_i = \check{\mathsf{cl}}p(\mathbf{0}). \tag{5}$$

Representing elements of the convex hull of $\mathcal{Y}_i \subseteq \mathbb{R}^{t+1}$ by Carathéodory theorem, we have that for each $i \in \overline{\mathcal{I}}$, there are vectors $\{\mathbf{u}_i^j\}_{j=1}^{t+2}$ and scalars $\{a_i^j\}_{j=1}^{t+2} \in \mathbb{R}$ such that

$$\sum_{j=1}^{t+2} a_i^j = 1, \quad a_i^j \geq 0, \ j \in [t+2],$$

$$\overline{\mathbf{r}}_i = \sum_{j=1}^{t+2} a_i^j \mathbf{u}_i^j =: \overline{\mathbf{u}}_i \in \text{dom}h_i, \qquad \overline{w}_i = \sum_{j=1}^{t+2} a_i^j h_i(\mathbf{u}_i^j). \tag{6}$$

Recall that we define

$$\check{\mathsf{cl}}h_i(\widetilde{\mathbf{u}}) := \inf_{\{a^j\}, \{\mathbf{u}_i^j\}} \left\{ \sum_{j=1}^{t+2} a^j h_i(\mathbf{u}_i^j) : \widetilde{\mathbf{u}} = \sum_{j=1}^{t+2} a^j \mathbf{u}_i^j, \sum_{j=1}^{t+2} a^j = 1, a^j \geq 0 \right\},$$

and $\Delta_\gamma^i := \sup_{\mathbf{u} \in \mathbb{R}^t} \{h_i(\mathbf{u}) - \check{\mathsf{cl}}h_i(\mathbf{u})\} \geq 0$. We have for $i \in \overline{\mathcal{I}}$,

$$\overline{w}_i \geq \check{\mathsf{cl}}h_i \left( \sum_{j=1}^{t+2} a_i^j \mathbf{u}_i^j \right) \quad \text{(by the definition of } \check{\mathsf{cl}}h_i(\cdot)\text{)}$$

$$\geq h_i \left( \sum_{j=1}^{t+2} a_i^j \mathbf{u}_i^j \right) - \Delta_\gamma^i \quad \text{(by the definition of } \Delta_\gamma^i\text{)} \tag{7}$$

$$= h_i(\overline{\mathbf{u}}_i) - \Delta_\gamma^i. \quad \text{(by Eqn. (6))}$$

Thus, by Eqns. (4) and (6), we have

$$\sum_{i=1}^{I} \overline{\mathbf{u}}_i = \mathbf{0}, \quad \overline{\mathbf{u}}_i \in \text{dom}h_i, \ i \in [I]. \tag{8}$$

Therefore, we have

$$
\begin{aligned}
p(\mathbf{0}) &= \sum_{i=1}^{I} h_i(\overline{\mathbf{u}}_i) \quad \text{(by Eqn. (8))} \\
&\leq \breve{\mathsf{cl}} p(\mathbf{0}) + \sum_{i \in \overline{\mathcal{I}}} \Delta_\gamma^i \quad \text{(by Eqns. (5) and (7))} \\
&\leq \breve{\mathsf{cl}} p(\mathbf{0}) + |\overline{\mathcal{I}}| \Delta_\gamma^{\mathsf{worst}} \\
&= \breve{\mathsf{cl}} p(\mathbf{0}) + (t+1) \Delta_\gamma^{\mathsf{worst}}, \quad \text{(by Lemma 6)}
\end{aligned}
$$

as desired. $\qquad\square$

By Lemmas 4 and 5, we have proved that

$$
\inf_{\gamma_1, \ldots, \gamma_I \in \mathbb{R}^g} \sup_{\theta \in \mathbb{R}^t} \frac{1}{I} \widetilde{\Phi}(\gamma_1, \ldots, \gamma_I; \theta) - \sup_{\theta \in \mathbb{R}^t} \inf_{\gamma_1, \ldots, \gamma_I \in \mathbb{R}^g} \frac{1}{I} \widetilde{\Phi}(\gamma_1, \ldots, \gamma_I; \theta) \leq \epsilon.
$$

To prove Theorem 1, we note that

$$
\begin{aligned}
w^* - q^* &:= \inf_{\gamma_1, \ldots, \gamma_I \in \mathbb{R}^g} \sup_{\theta \in \mathbb{R}^t} \frac{1}{I} \Phi(\gamma_1, \ldots, \gamma_I; \theta) - \sup_{\theta \in \mathbb{R}^t} \inf_{\gamma_1, \ldots, \gamma_I \in \mathbb{R}^g} \frac{1}{I} \Phi(\gamma_1, \ldots, \gamma_I; \theta) \\
&= \inf_{\gamma_1, \ldots, \gamma_I \in \mathbb{R}^g} \sup_{\theta \in \mathbb{R}^t} \frac{1}{I} \Phi(\gamma_1, \ldots, \gamma_I; \theta) - \inf_{\gamma_1, \ldots, \gamma_I \in \mathbb{R}^g} \sup_{\theta \in \mathbb{R}^t} \frac{1}{I} \widetilde{\Phi}(\gamma_1, \ldots, \gamma_I; \theta) \\
&\quad + \inf_{\gamma_1, \ldots, \gamma_I \in \mathbb{R}^g} \sup_{\theta \in \mathbb{R}^t} \frac{1}{I} \widetilde{\Phi}(\gamma_1, \ldots, \gamma_I; \theta) - \sup_{\theta \in \mathbb{R}^t} \inf_{\gamma_1, \ldots, \gamma_I \in \mathbb{R}^g} \frac{1}{I} \widetilde{\Phi}(\gamma_1, \ldots, \gamma_I; \theta) \\
&\quad + \sup_{\theta \in \mathbb{R}^t} \inf_{\gamma_1, \ldots, \gamma_I \in \mathbb{R}^g} \frac{1}{I} \widetilde{\Phi}(\gamma_1, \ldots, \gamma_I; \theta) - \sup_{\theta \in \mathbb{R}^t} \inf_{\gamma_1, \ldots, \gamma_I \in \mathbb{R}^g} \frac{1}{I} \Phi(\gamma_1, \ldots, \gamma_I; \theta) \\
&\leq \Delta_\theta^{\mathsf{minimax}} + \Delta_\theta^{\mathsf{maximin}} + \epsilon,
\end{aligned}
$$

as desired.

When $\phi(\gamma_i; \theta)$ is concave and closed w.r.t. discriminator parameter $\theta$, we have $\widehat{\mathsf{cl}}\phi = \phi$. Thus, $\Delta_\theta^{\mathsf{minimax}} = \Delta_\theta^{\mathsf{maximin}} = 0$ and $0 \leq w^* - q^* \leq \epsilon$.

## B.2 PROOFS OF THEOREM 3: EXISTENCE OF APPROXIMATE EQUILIBRIUM

We first show that the equilibrium value $V$ is $2f(1/2)$. For the discriminator $D_\theta$ which only outputs $1/2$, it has payoff $2f(1/2)$ for all possible implementations of generators $G_{\gamma_1}, \ldots, G_{\gamma_I}$. Therefore, we have $V \geq 2f(1/2)$. We now show that $V \leq 2f(1/2)$. We note that by assumption, for any $\xi > 0$, there exists a closed neighbour of implementation of generator $G_\xi$ such that $\mathbb{E}_{\mathbf{x} \sim \mathcal{P}_d, \mathbf{z} \sim \mathcal{P}_z} \|G'_\xi(\mathbf{z}) - \mathbf{x}\|_2 \leq \xi$ for all $G'_\xi$ in the neighbour. Such a neighbour exists because the generator is Lipschitz w.r.t. its parameters. Let the parameter implementation of such neighbour of $G_\xi$ be $\Gamma$. The Wasserstein distance between $G_\xi$ and $\mathcal{P}_d$ is $\xi$. Since the function $f$ and the discriminator are $L_f$-Lipschitz and $L$-Lipschitz, respectively, we have

$$
\left| \mathbb{E}_{\mathbf{z} \sim G_\xi} f(1 - D_\theta(\mathbf{z})) - \mathbb{E}_{\mathbf{x} \sim \mathcal{P}_d} f(1 - D_\theta(\mathbf{x})) \right| \leq \mathcal{O}(L_f L \xi).
$$

Thus, for any fixed $\gamma$, we have

$$
\begin{aligned}
&\sup_{\theta \in \mathbb{R}^t} \mathbb{E}_{\mathbf{x} \sim \mathcal{P}_d} f(D_\theta(\mathbf{x})) + \mathbb{E}_{\mathbf{z} \sim G_\xi} f(1 - D_\theta(\mathbf{z})) \\
&\leq \mathcal{O}(L_f L \xi) + \sup_{\theta \in \mathbb{R}^t} \mathbb{E}_{\mathbf{x} \sim \mathcal{P}_d} f(D_\theta(\mathbf{x})) + \mathbb{E}_{\mathbf{x} \sim \mathcal{P}_d} f(1 - D_\theta(\mathbf{x})) \\
&\leq \mathcal{O}(L_f L \xi) + 2f(1/2) \to 2f(1/2), \quad (\xi \to +0)
\end{aligned}
$$

which implies that $\sup_{\theta \in \mathbb{R}^t} \Phi(\gamma_1, \ldots, \gamma_I; \theta) = 2f(1/2)$ for all $\gamma_1, \ldots, \gamma_I \in \Gamma$. So we have $V = 2f(1/2)$. This means that the discriminator cannot do much better than a random guess.

The above analysis implies that the equilibrium is achieved when $D_{\theta^*}$ only outputs $1/2$. Denote by $\Theta$ the small closed neighbour of such $\theta^*$ such that $\Phi(\gamma_1, \ldots, \gamma_I; \theta)$ is concave w.r.t. $\theta \in \Theta$ for any fixed $\gamma_1, \ldots, \gamma_I \in \Gamma$. We thus focus on the loss on $\Theta \subseteq \mathbb{R}^t$ and $\Gamma \subseteq \mathbb{R}^g$:

$$
\Phi(\gamma_1, \ldots, \gamma_I; \theta) := \sum_{i=1}^{I} \left[ \mathbb{E}_{\mathbf{x} \sim \mathcal{P}_d} f(D_\theta(\mathbf{x})) + \mathbb{E}_{\mathbf{z} \sim \mathcal{P}_z} f(1 - D_\theta(G_{\gamma_i}(\mathbf{z}))) \right], \quad \theta \in \Theta, \ \gamma_1, \ldots, \gamma_I \in \Gamma.
$$

Since $\Phi(\gamma_1, ..., \gamma_I; \theta)$ is concave w.r.t. $\theta \in \Theta$ for all $\gamma_1, ..., \gamma_I \in \Gamma$, by Corollary 2, we have

$$\inf_{\gamma_1,...,\gamma_I \in \Gamma} \sup_{\theta \in \Theta} \frac{1}{I}\Phi(\gamma_1, ..., \gamma_I; \theta) - \sup_{\theta \in \Theta} \inf_{\gamma_1,...,\gamma_I \in \Gamma} \frac{1}{I}\Phi(\gamma_1, ..., \gamma_I; \theta) \leq \epsilon.$$

The optimal implementations of $\gamma_1, ..., \gamma_I$ is achieved by $\operatorname{argmin}_{\gamma_1,...,\gamma_I \in \Gamma} \sup_{\theta \in \Theta} \frac{1}{I}\Phi(\gamma_1, ..., \gamma_I; \theta)$.

## C  USEFUL LEMMAS

**Lemma 7.** *Given the function*

$$(f_1 + ... + f_I)(\theta) := f_1(\theta) + ... + f_I(\theta),$$

*where $f_i : \mathbb{R}^t \to \mathbb{R}$, $i \in [I]$ are closed proper convex functions. Denote by $f_1^* \oplus ... \oplus f_I^*$ the infimal convolution*

$$(f_1^* \oplus ... \oplus f_I^*)(\mathbf{u}) := \inf_{\mathbf{u}_1 + ... + \mathbf{u}_I = \mathbf{u}} \{f_1^*(\mathbf{u}_1) + ... + f_I^*(\mathbf{u}_I)\}, \qquad \mathbf{u} \in \mathbb{R}^t.$$

*Provided that $f_1 + ... + f_I$ is proper, then we have*

$$(f_1 + ... + f_I)^*(\mathbf{u}) = \mathsf{cl}(f_1^* \oplus ... \oplus f_I^*)(\mathbf{u}), \quad \forall \mathbf{u} \in \mathbb{R}^t.$$

*Proof.* For all $\theta \in \mathbb{R}^t$, we have

$$
\begin{aligned}
f_1(\theta) + ... + f_I(\theta) &= \sup_{\mathbf{u}_1}\{\theta^T\mathbf{u}_1 - f_1^*(\mathbf{u}_1)\} + ... + \sup_{\mathbf{u}_I}\{\theta^T\mathbf{u}_I - f_I^*(\mathbf{u}_I)\} \\
&= \sup_{\mathbf{u}_1,...,\mathbf{u}_I}\{\theta^T(\mathbf{u}_1 + ... + \mathbf{u}_I) - f_1^*(\mathbf{u}_1) - ... - f_I^*(\mathbf{u}_I)\} \\
&= \sup_{\mathbf{u}} \sup_{\mathbf{u}_1 + ... + \mathbf{u}_I = \mathbf{u}}\{\theta^T\mathbf{u} - f_1^*(\mathbf{u}_1) - ... - f_I^*(\mathbf{u}_I)\} \\
&= \sup_{\mathbf{u}}\left\{\theta^T\mathbf{u} - \inf_{\mathbf{u}_1 + ... + \mathbf{u}_I = \mathbf{u}} f_1^*(\mathbf{u}_1) - ... - f_I^*(\mathbf{u}_I)\right\} \\
&= \sup_{\mathbf{u}}\{\theta^T\mathbf{u} - (f_1^* \oplus ... \oplus f_I^*)(\mathbf{u})\} \\
&= (f_1^* \oplus ... \oplus f_I^*)^*(\theta).
\end{aligned}
\tag{9}
$$

Therefore,

$$\mathsf{cl}(f_1^* \oplus ... \oplus f_I^*)(\mathbf{u}) = \check{\mathsf{cl}}(f_1^* \oplus ... \oplus f_I^*)(\mathbf{u}) = (f_1^* \oplus ... \oplus f_I^*)^{**}(\mathbf{u}) = (f_1 + ... + f_I)^*(\mathbf{u}),$$

where the first equality holds because $(f_1^* \oplus ... \oplus f_I^*)$ is convex, the second quality is by standard conjugate theorem, and the last equality holds by conjugating the both sides of Eqn. (9).  □

**Lemma 8** (Proposition 3.4 (b), Bertsekas (2009))**.** *For any function $p(\mathbf{u})$, denote by $q(\mu) := \inf_{\mathbf{u} \in \mathbb{R}^t}\{p(\mathbf{u}) + \mu^T\mathbf{u}\}$. We have $\sup_{\mu \in \mathbb{R}^t} q(\mu) = \check{\mathsf{cl}}p(\mathbf{0})$.*

## D  DISTRIBUTIONAL APPROXIMATION PROPERTIES OF STACKELBERG GAN

**Theorem 9.** *Suppose that $f$ is strictly concave and the discriminator has infinite capacity. Then, the global optimum of Stackelberg GAN is achieved if and only if*

$$\frac{1}{I}\sum_{i=1}^{I} \mathcal{P}_{G_{\gamma_i}(\mathbf{z})} = \mathcal{P}_d.$$

*Proof.* We define

$$L(\mathcal{P}_d, \mathcal{P}_{G_{\gamma}(\mathbf{z})}) = \sup_{\theta \in \mathbb{R}^t} \mathbb{E}_{\mathbf{x} \sim \mathcal{P}_d} f(D_\theta(\mathbf{x})) + \mathbb{E}_{\mathbf{z} \sim \mathcal{P}_z} f(1 - D_\theta(G_\gamma(\mathbf{z}))).$$

Clearly, the vanilla GAN optimization can be understood as projecting under $L$:

$$\inf_{\gamma \in \mathbb{R}^g} L(\mathcal{P}_d, \mathcal{P}_{G_{\gamma}(\mathbf{z})}).$$

In the Stackelberg GAN setting, we are projecting under a different distance $\tilde{L}$ which is defined as

$$\tilde{L} = \sup_{\theta \in \mathbb{R}^t} \mathbb{E}_{\mathbf{x} \sim \mathcal{P}_d} f(D_\theta(\mathbf{x})) + \frac{1}{I} \sum_{i=1}^{I} \mathbb{E}_{\mathbf{z} \sim \mathcal{P}_z} f(1 - D_\theta(G_{\gamma_i}(\mathbf{z}))) \tag{10}$$

$$= L\left(\mathcal{P}_d, \frac{1}{I} \sum_{i=1}^{I} \mathcal{P}_{G_{\gamma_i}(\mathbf{z})}\right). \tag{11}$$

We note that $f$ is strictly concave and the discriminator has capacity large enough implies the followings: $L(\mathcal{P}_1, \mathcal{P}_2)$, as a function of $\mathcal{P}_2$, achieves the global minimum if and only if $\mathcal{P}_2 = \mathcal{P}_1$. The theorem then follows from this fact and (11).

$\square$

## E    NETWORK SETUP

Table 2: Architecture and hyper-parameters for the mixture of Gaussians dataset.

| | Operation | Input Dim | Output Dim | BN? | Activation |
|---|---|---|---|---|---|
| Generator $G(\mathbf{z}) : \mathbf{z} \sim \mathcal{N}(0, 1)$ | | | 2 | | |
| | Linear | 2 | 16 | ✓ | |
| | Linear | 16 | 2 | | Tanh |
| Discriminator | | | | | |
| | Linear | 2 | 512 | | Leaky ReLU |
| | Linear | 512 | 256 | | Leaky ReLU |
| | Linear | 256 | 1 | | Sigmoid |
| Number of generators | 8 | | | | |
| Batch size for real data | 64 | | | | |
| Number of iterations | 200 | | | | |
| Slope of Leaky ReLU | 0.2 | | | | |
| Learning rate | 0.0002 | | | | |
| Optimizer | Adam | | | | |

Table 3: Architecture and hyper-parameters for the MNIST datasets.

| | Operation | Input Dim | Output Dim | BN? | Activation |
|---|---|---|---|---|---|
| Generator $G(\mathbf{z}) : \mathbf{z} \sim \mathcal{N}(0, 1)$ | | | 2 | | |
| | Linear | 100 | 512 | ✓ | |
| | Linear | 512 | 784 | | Tanh |
| Discriminator | | | | | |
| | Linear | 2 | 512 | | Leaky ReLU |
| | Linear | 512 | 256 | | Leaky ReLU |
| | Linear | 256 | 1 | | Sigmoid |
| Number of generators | 10 | | | | |
| Batch size for real data | 100 | | | | |
| Slope of Leaky ReLU | 0.2 | | | | |
| Learning rate | 0.0002 | | | | |
| Optimizer | Adam | | | | |

Table 4: Architecture and hyper-parameters for the Fashion-MNIST datasets.

| | Operation | Input Dim | Output Dim | BN? | Activation |
|---|---|---|---|---|---|
| Generator $G(\mathbf{z}) : \mathbf{z} \sim \mathcal{N}(\mathbf{0}, \mathbf{1})$ | | | 2 | | |
| | Linear | 2 | 128 | ✓ | |
| | Linear | 128 | 256 | ✓ | |
| | Linear | 256 | 512 | ✓ | |
| | Linear | 512 | 1024 | ✓ | |
| | Linear | 1024 | 784 | | Tanh |
| Discriminator | | | | | |
| | Linear | 2 | 512 | | Leaky ReLU |
| | Linear | 512 | 256 | | Leaky ReLU |
| | Linear | 256 | 1 | | Sigmoid |
| Number of generators | 10 | | | | |
| Batch size for real data | 100 | | | | |
| Number of iterations | 500 | | | | |
| Slope of Leaky ReLU | 0.2 | | | | |
| Learning rate | 0.0002 | | | | |
| Optimizer | Adam | | | | |

Table 5: Architecture and hyper-parameters for the CIFAR-10 dataset.

| | Operation | Kernel | Strides | Feature maps | BN? | BN center? | Activation |
|---|---|---|---|---|---|---|---|
| $G(\mathbf{z}) : \mathbf{z} \sim \mathsf{Uniform}[-1, 1]$ | | | | 100 | | | |
| | Fully connected | | | 8×8×128 | ✗ | ✗ | ReLU |
| | Transposed convolution | 5×5 | 2×2 | 64 | ✗ | ✗ | ReLU |
| | Transposed convolution | 5×5 | 2×2 | 3 | ✗ | ✗ | Tanh |
| $D(\mathbf{x})$ | | | | 8×8×256 | | | |
| | Convolution | 5×5 | 2×2 | 128 | ✓ | ✓ | Leaky ReLU |
| | Convolution | 5×5 | 2×2 | 256 | ✓ | ✓ | Leaky ReLU |
| | Convolution | 5×5 | 2×2 | 512 | ✓ | ✓ | Leaky ReLU |
| | Fully connected | | | 1 | ✗ | ✗ | Sigmoid |
| Number of generators | 10 | | | | | | |
| Batch size for real data | 64 | | | | | | |
| Batch size for each generator | 64 | | | | | | |
| Number of iterations | 100 | | | | | | |
| Slope of Leaky ReLU | 0.2 | | | | | | |
| Learning rate | 0.0002 | | | | | | |
| Optimizer | Adam($\beta_1 = 0.5, \beta_2 = 0.999$) | | | | | | |
| Weight, bias initialization | $\mathcal{N}(\mu = 0, \sigma = 0.01), 0$ | | | | | | |

Table 6: Architecture and hyper-parameters for the Tiny ImageNet dataset.

| Operation | Kernel | Strides | Feature maps | BN? | BN center? | Activation |
|---|---|---|---|---|---|---|
| $G(\mathbf{z}) : \mathbf{z} \sim \mathsf{Uniform}[-1, 1]$ | | | 100 | | | |
| Fully connected | | | 8×8×256 | ✗ | ✗ | ReLU |
| Transposed convolution | 5×5 | 2×2 | 128 | ✗ | ✗ | ReLU |
| Transposed convolution | 5×5 | 2×2 | 3 | ✗ | ✗ | Tanh |
| $D(\mathbf{x})$ | | | 8×8×256 | | | |
| Convolution | 5×5 | 2×2 | 128 | ✓ | ✓ | Leaky ReLU |
| Convolution | 5×5 | 2×2 | 256 | ✓ | ✓ | Leaky ReLU |
| Convolution | 5×5 | 2×2 | 512 | ✓ | ✓ | Leaky ReLU |
| Fully connected | | | 1 | ✗ | ✗ | Sigmoid |
| Number of generators | 10 | | | | | |
| Batch size for real data | 64 | | | | | |
| Batch size for each generator | 64 | | | | | |
| Number of iterations | 300 | | | | | |
| Slope of Leaky ReLU | 0.2 | | | | | |
| Learning rate | 0.00001 | | | | | |
| Optimizer | $\mathsf{Adam}(\beta_1 = 0.5, \beta_2 = 0.999)$ | | | | | |
| Weight, bias initialization | $\mathcal{N}(\mu = 0, \sigma = 0.01), 0$ | | | | | |

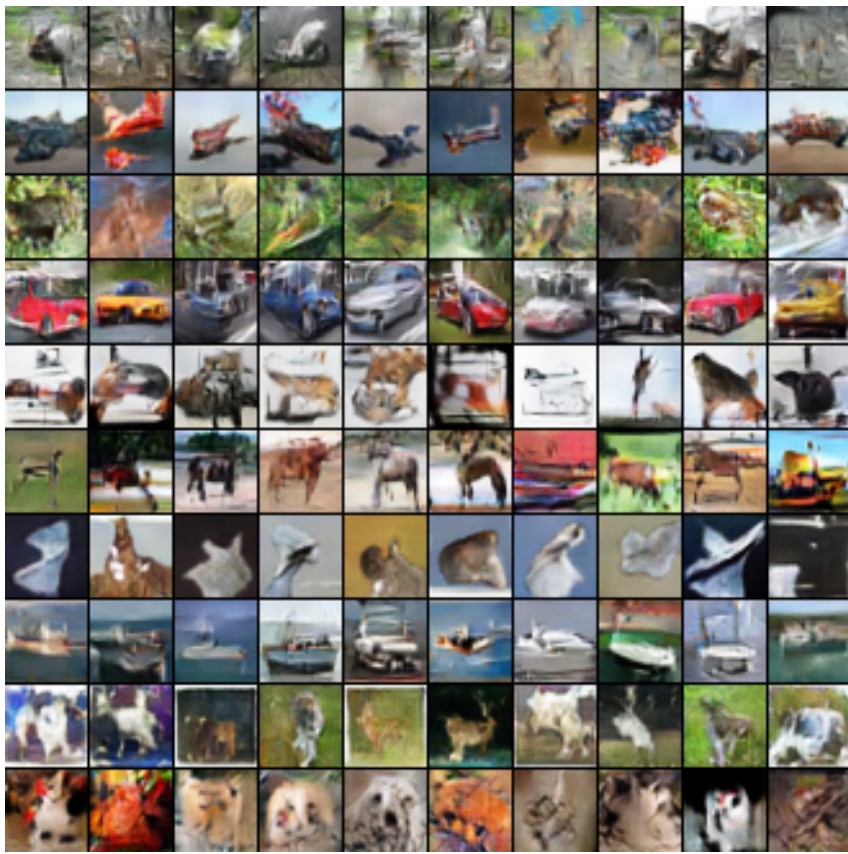

Figure 10: Examples generated by Stackelberg GAN with 10 generators on CIFAR-10 dataset, where each row corresponds to samples from one generator.

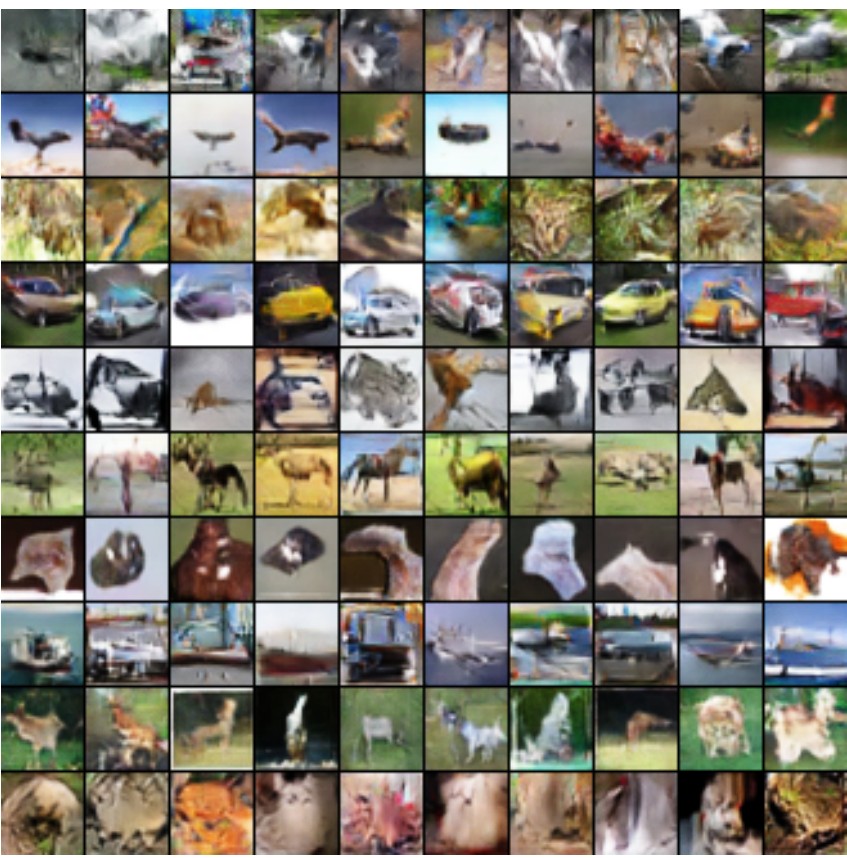

Figure 11: Examples generated by Stackelberg GAN with 10 generators on CIFAR-10 dataset, where each row corresponds to samples from one generator.

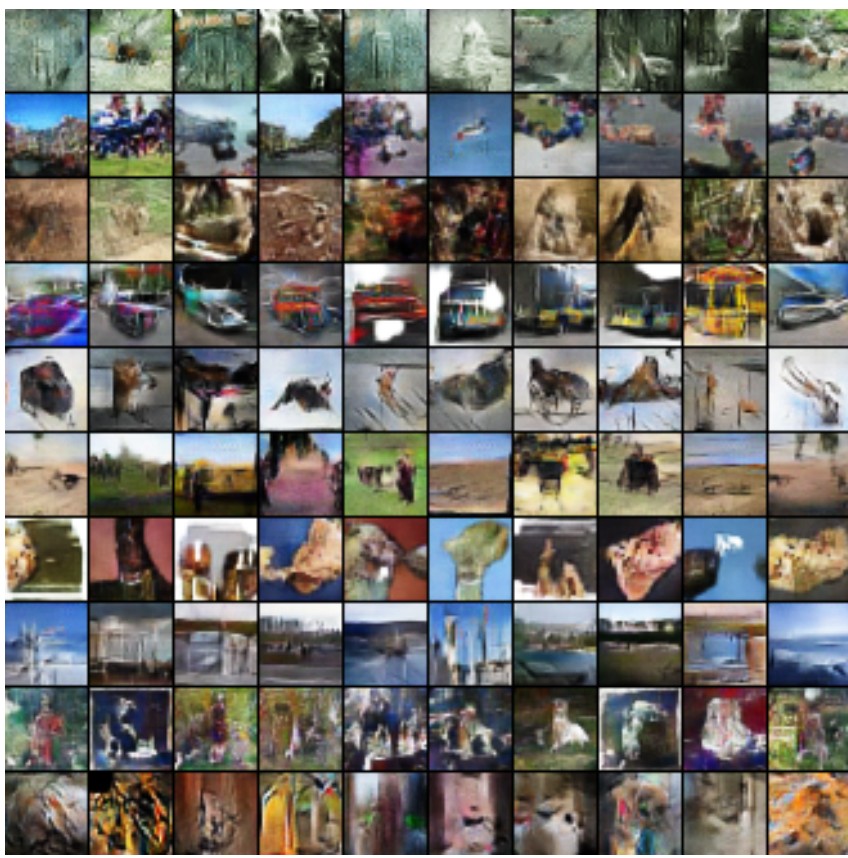

Figure 12: Examples generated by Stackelberg GAN with 10 generators on Tiny ImageNet dataset, where each row corresponds to samples from one generator.

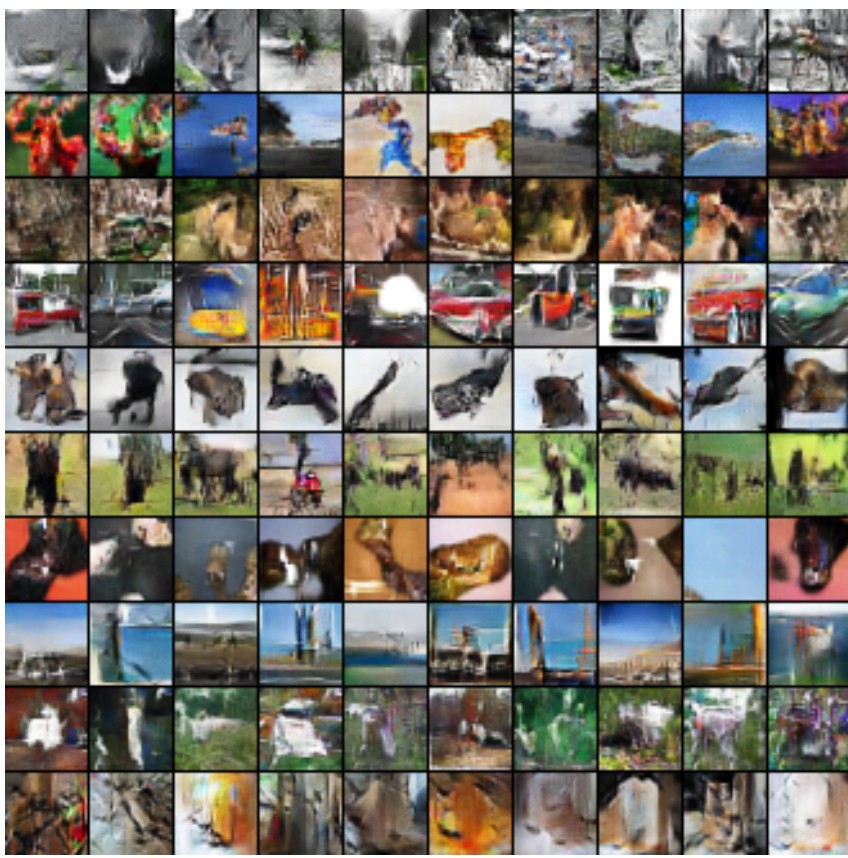

Figure 13: Examples generated by Stackelberg GAN with 10 generators on Tiny ImageNet dataset, where each row corresponds to samples from one generator.

