# OpenReview forum: "Stackelberg GAN: Towards Provable Minimax Equilibrium via Multi-Generator Architectures"
_ICLR.cc/2019/Conference_

### Official Review · AnonReviewer1 · 2018-11-02
**The name Stackelberg GAN is misleading as the underlying problem formulation is not a Stackelberg game but (still) a zero-sum game. The argument why more data generators help is not convincing.**

**Rating:** 4
**Confidence:** 4

**Review:**

A Stackelberg competition is a nonzero-sum game where 1) each player has their own objective, which do not sum up to a constant, and 2) there is an order at which the players interact. The proposed formulation only assumes that parameters of one player (data generator) partition in I tuples \gamma_i of parameters, where each tuple parameterizes a different data generator component (e.g., a separate neural network). Further, each of those components is assumed to contribute a term to the game's objective that only depends on the corresponding parameter tuple \gamma_i, and the other player's parameters \theta (e.g., weights of the discriminator). From a game theoretic perspective, this still yields a 2-player zero-sum game where the action space of the data generator is the product space of the I tuple spaces. Hence, I have doubts about the general finding that more data generating components decreases the duality gap.

The gap between the a maximin and minimax solution is determined by the shape of the objective \phi(\gamma,\theta) and is zero, for example, if \phi is (quasi) convex in \gamma=[\gamma_1, ..., \gamma_I], and (quasi) concave in \theta. The authors bound the violation of this property w.r.t. the data generator components' parameters \gamma_i, and argue that this degree of violation is the same for the whole data generator parametrized by \gamma=[\gamma_1, ..., \gamma_I] if the data generator components are from the same family of mappings (e.g., having the same network architecture). While this conclusion is true under worst cast assumption, e.g., the globally maximal possible gap, this would also imply that all data generator components find the same global best solution, that is, yield the same mapping, in which case the gap would be identical to just using one of those components.

Intuitively, the only reason to have multiple data generator components is to learn different mappings such that the joint data generator -- mixing the outputs of the different components -- is more expressiv than just a single mapping. If the different mappings only result from the inability of finding the global best solution, a worst case argument is not very insightful; in this case, one should study the duality gap in the neighborhood of the starting solutions. On the other hand, if we assume a different family of mappings for each component, the convexity violation of the joint data generator is higher than for each component; hence, the gap does not necessarily decrease with more components.

So why do multiple data generator components help in practice, and why does the proposed model outperform single-component GANs and the multi-branch GAN in the experiments? Solving a maximin/minimax problem for highly non-convex-concave functions is challenging; there is an infinity of saddle point solutions which yield different "performances". The multi-branch GAN can be seen as a model averaging approach giving more stable results, whereas the proposed GAN seems more of an ensemble approach to stabilize the result. Though, this is speculative and I would encourage the authors to study this in-depth; the reasoning in Remark 1 is not convincing to me.

UPDATE:

I read the revision and stick to my vote. In the discussion, I wasn't able to get my points across, e.g., that bounding the worst case duality gap is not enough to conclude that the observed duality gap does not grow for multiple local optimal GANs, where the duality gap is expected to be much smaller. A simple experiment could be to actually measure the duality gap (flip the order of the players and measure the difference of the objectives, when starting with the same initialization). If the authors were right, the maximum of those gap should stay constant when adding more data generators. To justify a Stackelberg setting, the authors may provide an example instantiation that cannot be cast into a standard zero-sum game with minimax solution. I can't see such an example but I'm happy to be proven wrong.

---

> ### Author Response · Authors · 2018-11-09
> **Stackelberg game can be a zero-sum game. We provide a necessary condition about why more data generators help by analyzing the existence of approximate Nash equilibrium.**
>
> We thank the reviewer for the comments. However, we respectfully disagree with a few points from the reviewer. We will address all concerns from the reviewer in the following form of Q/A.
>
> Q1 (The first paragraph of review). A Stackelberg competition is a non-zero-sum game. From a game theoretic perspective, Stackelberg GAN still yields a 2-player zero-sum game. Hence, I have doubts about the general finding that more data generating components decreases the duality gap.
> A1. Stackelberg competition can be a zero-sum game. There are many references supporting this claim. We cite some of them below. In fact, zero-sum Stackelberg games are equivalent to solving for the minimax equilibrium in zero-sum games. So, people usually don't talk about Stackelberg equilibrium in zero-sum games, instead they talk about minimax equilibrium. Here we mainly use the concept of leader-follower model in stackelberg game to represent the sequential adversarial process between one discriminator and multiple generators. Therefore, we respectfully disagree that the name Stackelberg GAN is misleading, since the underlying problem formulation is indeed a zero-sum Stackelberg game.
> [1] Stackelberg Security Games: Looking Beyond a Decade of Success, 2018.
> [2] http://coral.ise.lehigh.edu/wp-content/uploads/coralseminar/ipsem/talks/2005_06/scott_bicrit.pdf
> [3] Sequential Stackelberg Equilibria in Two-Person Games, 1988.
>
> Q2 (The second paragraph of review). Suppose that the data generator components have the same network architecture. This would also imply that all data generator components find the same global best solution, in which case the gap would be identical to just using one of those components.
> A2. We argue that having the same network architecture for each data generator component does not necessarily imply all data generator components find the same global best solution. Here we provide two reasons about this. (1) Neural network is highly non-convex: starting from different random initializations, each generator would converge to different solutions even with the same network architecture, as the globally optimal solutions might not be unique. (2) In Appendix D, we show that Stackelberg GAN can learn a mixture of distributions under standard assumptions as Goodfellow et al.’14. This implies that all generators cannot find the same globally best solution when each generator does not have enough capacity to learn the real data distribution but a mixture of generators has; otherwise, we have P_{G_{gamma_i}(z)}=P_d for all i, contradicting with the condition that “each generator does not have enough capacity to learn the real data distribution”. From (1) and (2), we see that “different initializations” and “generator capacity” are two factors which might prevent generators from finding the same solution.
> Our main conclusion holds not only for the worst case, but also holds true for the practical cases. For example, in Figure 1 we use the same network architecture for all generator components. We do not observe the phenomenon that all data generator components find the same globally best solution as the reviewer mentioned.
>
> Q3 (The third paragraph of review). If we assume a different family of mappings for each component, the convexity violation of the joint data generator is higher than for each component; hence, the gap does not necessarily decrease with more components.
> A3. We respectfully disagree with the comment. Denote by Delta_i the convexity violation of the i-th generator and let Delta_max=max{Delta_1,…,Delta_I}. Our result shows that the convexity violation of the joint data generator (i.e., the duality gap) is no larger than Delta_max/I. Since Delta_max can be a bounded value, this shows that the gap decreases with more components. Indeed, the convexity violation of the joint data generator Delta_max/I is smaller than that of the most non-convex generator, and can even be smaller than the most convex generator when I is sufficiently large.
>
> Q4 (The fourth paragraph of review) So why do multiple data generator components help in practice, and why does the proposed model outperform single-component GANs and the multi-branch GAN in the experiments?
> A4. We answer both questions from the game-theoretic perspective in this paper --- when does approximate equilibrium exist. We believe both points of view (optimization and game theory) are worth studying, while we focus on the latter one. We argue that there are strong connections between the two points of view: if approximate Nash equilibrium does not exist as in the single-component GANs, all optimization methods would suffer from instability and finally fail. In contrast, our study shows that approximate Nash equilibrium exists for multi-component GANs and improves over Arora’s result. So, our study of Nash equilibrium serves as a necessary condition for the success of GANs.
>
> We are looking forward to your re-evaluation based on our reply. Thanks for your consideration.

---

### Official Review · AnonReviewer3 · 2018-11-03
**A nice approach for training multi-generator**

**Rating:** 7
**Confidence:** 3

**Review:**

This paper proposes a way of training multi-generator in the GAN setting.
While a proposed approach is simply to put N generators and form a sum of GAN losses to train a model, the paper carefully presents a theoretical analysis on the duality gap, and shows as N goes infinity, the duality gap can shrink to zero.
One can think of this as a usual ensemble approach to increase model's capacity and performance, but the main difference to the usual ensemble approach is to form a sum of losses (ensemble losses) instead of a loss on output of ensemble.
The paper shows this can be more effective approach to train a multi-generator architecture and I believe that this can be an effective approach to capture multi-modal sample distributions.
Finally, a paper is well-written and well-organized.

---

> ### Comment · AnonReviewer2 · 2018-11-03
> **The difference between a sum of losses (ensemble losses) and a loss on output of ensemble is very minor, even negligible**
>
> The authors overstate the difference between a sum of losses (ensemble losses) and a loss on output of ensemble. In fact, in terms of algorithm, this difference is very minor, even negligible.
>
> When we have a loss on output of ensemble, the loss is:
> \sum_{n=1}^N f(D_{\theta}(x_n) + \sum_{i=1}^I \sum_{n=1}^{N_i} f(1-D_{\theta}(G_{\gamma_i}(z_{i,n}))),
> where {x_n}_{n=1}^N are iid samples from the dataset, {G_{\gamma}(z_{n,i}): 1\le i\le I, 1\le n \le N_i} are iid samples from the mixture generative model, and \sum_{i=1}^I N_i = N.
>
> When we have a sum of losses (ensemble losses), the loss is:
> \sum_{n=1}^N f(D_{\theta}(x_n) + \sum_{i=1}^I \sum_{n=1}^{N/I} f(1-D_{\theta}(G_{\gamma_i}(z_{i,n}))),
> where {x_n}_{n=1}^N are iid samples from the dataset, {G_{\gamma}(z_{n,i}): 1\le n \le N/I} are iid samples from the i'th generative model, and all the I generator components contribute N/I samples to the loss equally.
>
> Therefore, the only difference is that in a loss on output of ensemble, we are truly sampling from the ensemble model, while in a sum of losses (ensemble losses), we enforce that each component must contribute the same number of training samples.
>
> This difference even makes Stackelberg GAN more difficult to prove its convergence in the density approximation sense, because now its training samples are not iid sampled from the ensemble model.

---

> > ### Author Response · Authors · 2018-11-06
> > **Misunderstandings about the ensemble loss by AnonReviewer2**
> >
> > We thank AnonReviewer2 for the comment. We believe there are misunderstandings about the loss by the reviewer here. Our ensemble loss is E_x f(D_{\theta}(x)) + \frac{1}{I}\sum_{i=1}^I E_z f(1-D_{\theta}(G_{\gamma_i}(z))). This is totally different from the loss that the reviewer mentioned, as the index i is imposed on the generator parameter \gamma in our loss. Our loss involves optimizing *multiple* generators jointly, while the ensemble loss that the reviewer mentioned only involves learning one generator. Therefore, there is huge difference between the ensemble loss (as in this paper) and the loss on output of ensemble (as the reviewer mentioned).
> >
> > Furthermore, we do not require that each component must contribute the same number of training samples in the ensemble loss. Rather, we only restrict the *weight* of all generators to be the same. Our analysis focuses on the population form where many sampling methods are consistent with it by the law of large number. For example, we allow the generator mixture model with uniform distribution over all generators. We also allow an empirical ensemble loss \frac{1}{N}\sum_{n=1}^N f(D_{\theta}(x_n)) + \frac{1}{I}\sum_{i=1}^I \frac{1}{N_i}\sum_{n=1}^{N_i} f(1-D_{\theta}(G_{\gamma_i}(z_{i,n}))) with i.i.d. z_{i,n}. We even allow the case that N_1=…=N_I. So, our model does not have the issue that “Stackelberg GAN is more difficult to prove its convergence in the density approximation sense”, since the training samples indeed can be i.i.d. sampled from the ensemble model.
> >
> > Thanks again for your revaluation.

---

> > > ### Comment · AnonReviewer2 · 2018-11-06
> > > **The difference is in the empirical ensemble loss**
> > >
> > > Thanks for the reply!
> > >
> > > First, sorry that in my empirical ensemble losses I missed typed G_{\gamma_i} as G_{\gamma}. Both the sum of losses (ensemble losses) and a loss on output of ensemble have G_{\gamma_i}. After I corrected my typo, I did not see much difference there.
> > >
> > > In fact, if we consider the population loss, the sum of losses (ensemble losses) and the loss on output of ensemble are exactly the same. Both of them are E_x f(D_{\theta}(x)) + \frac{1}{I}\sum_{i=1}^I E_z f(1-D_{\theta}(G_{\gamma_i}(z))). The loss on output of ensemble can have different weights on different generators, and you method can do it, too.
> > >
> > > The difference is in the empirical ensemble loss, as I wrote in my last post. In the loss on output of ensemble, (N_1, N_2, ..., N_I) is a random vector with multinomial distribution (N, 1/I, 1/I, ..., 1/I). It is not clear to me that in your sum of losses (ensemble losses), how will you choose your (N_1, N_2, ..., N_I)? It seems to be that you choose N_1=…=N_I=N/I?

---

> > > > ### Author Response · Authors · 2018-11-06
> > > > **We do not need to choose N_1=…=N_I=N/I, although we can as well. The choice is flexible.**
> > > >
> > > > We thank AnonReviewer2 for the quick reply! Yes, the difference is in the empirical loss. However, we do not necessarily need to choose N_1=…=N_I=N/I, although we can as well. We believe the key to your question is on the relationship between population loss and empirical loss --- the unbiased estimator. Note that by the uniform convergence, an unbiased empirical loss asymptotically converges to the population loss. There are multiple ways of samplings which lead to an unbiased empirical loss to the reviewer’s population loss. Here are three examples: (1) the multinomial distribution with parameter (1/I,…,1/I) as the reviewer mentioned. Note that even in this case, with high probability N_1=…=N_I=N/I does not hold. (2) Each generator samples a fixed but unequal number of data points independently, e.g., N_1=1.5N/I, N_2=…=N_{I-1}=N/I, N_I=0.5/I. (3) Each generator samples a fixed and equal number of data points independently, i.e., N_1=…=N/I. All the three sampling schemes are unbiased to the population loss, although N_1=…=N_I=N/I does not always hold true.
> > > >
> > > > Thanks again for your question.

---

> > > > > ### Comment · AnonReviewer3 · 2018-12-01
> > > > > **I don't see why the review2 says the paper's contribution is negligible.**
> > > > >
> > > > > The paper's treatment clearly changes the order of taking expectation, i.e.
> > > > > minimizing the average of loss vs. minimizing the loss of averages.
> > > > >
> > > > > If the problem has a linear structure, then it's true that changing the order does not matter, but in this case, it does matter, and the paper supports this claim by showing experimental results. (I'm pretty sure depending on the problem structure one is the upper/lower bound to the other in certain cases.)

---

> > > > > > ### Comment · AnonReviewer2 · 2018-12-07
> > > > > > **changing the order of expectations v.s. explicitly computing one expectation**
> > > > > >
> > > > > > Hi Reviewer 3,
> > > > > >
> > > > > > From my point of view, the proposed method does not change the order of expectations, but just explicitly computing the expectation with respect to the multinomial distribution (N, 1/I, 1/I, ..., 1/I) over the N discriminators.
> > > > > >
> > > > > > When we have a loss on output of ensemble, the empirical loss is:
> > > > > > \sum_{n=1}^N f(D_{\theta}(x_n) + \sum_{i=1}^I \sum_{n=1}^{N_i} f(1-D_{\theta}(G_{\gamma_i}(z_{i,n}))),
> > > > > > where {x_n}_{n=1}^N are iid samples from the dataset, {G_{\gamma}(z_{n,i}): 1\le i\le I, 1\le n \le N_i} are iid samples from the mixture generative model, and \sum_{i=1}^I N_i = N.
> > > > > >
> > > > > > The population loss is
> > > > > > \Expect_{x_n}[ \sum_{n=1}^N f(D_{\theta}(x_n)) ] + \Expect_{N_i, z_{i,n}}[ \sum_{i=1}^I \sum_{n=1}^{N_i} f(1-D_{\theta}(G_{\gamma_i}(z_{i,n}))) ].
> > > > > > The first expectation is easy, resulting in N \Expect_{x} [ f(D_{\theta}(x)) ].
> > > > > > The second expectation can be simplified by performing a conditional expectation on N_i, and then explicitly computing the expectation with respect to N_i. (I omit a few steps here, but I can provide if needed.) This results in
> > > > > > \sum_{i=1}^I N/I \Expect_{z_i}[ f(1-D_{\theta}(G_{\gamma_i}(z_{i}))) ],
> > > > > > which is exactly the population loss proposed in this paper.
> > > > > >
> > > > > > Therefore, I think that there's no exchange of expectations in the proposed method. The population loss function used in this paper and in previous ensemble-discriminator papers are the same. The difference is only in the empirical loss, which I think is minor.

---

### Official Review · AnonReviewer2 · 2018-11-03
**Interesting view of GANs from game-theory perspective, but algorithmically the Stackelberg GAN is similar with previous multiple-generator GANs**

**Rating:** 5
**Confidence:** 3

**Review:**

This paper proposes the Stackelberg GAN framework of multiple generators in the GAN architecture. The architecture is similar with previous multiple-generator GANs (MAD-GAN and MGAN). In fact, it's even simpler in the sense that Stackelberg GAN has simpler loss function for the discriminator compared with the previous two. The authors prove that the minimax duality gap shrinks as the number of generators increases. And this proof has no assumption on the expressive power of generators and discriminator. With this proof, the authors argues that because the duality gap shrinks as the number of generators increases, the training of GANs gets more stable.

From the algorithm part, I think the algorithm is very similar (and even simpler) than MAD-GAN and MGAN. The MAD-GAN and MGAN even proposed some specific loss for the discriminator so that it will encourage different generator to generate different modes in the target distribution. The Stackelberg GAN does not do this, but "partially" achieved the same goal. However, from Figure 9, we see that the simpler the generator is, the easier different generator will capture different modes. I think that this is due to the simplicity of discriminator loss. Therefore, on the algorithm part, the author may want to address the difference between Stackelberg GAN and MAD-GAN and MGAN. On the experiment part, we need to see more comparison between these three methods. In the current experiment, MGAN result is very similar to the proposed method, and MAD-GAN result is missing. Personally, I think that on cifar dataset (or larger datasets), these three methods should have very similar behavior.

From the theoretical part, the authors derived a bound of the minimax duality gap for the Stackelberg GAN, without the assumption on the expressive power of generators and discriminator. Although the bound may not be practical, these are nice efforts. There are many typos in the paper (and appendix), which make me difficult to follow the proofs. For example, "Let clf (bclf) be the convex(concave) closure of f, which is defined as the function whose epigraph (subgraph) is the convex
(concave) closed hull of that of function f." Do we have concave closed hull of subgraph of function f? What is the concave closed hull of a set? The usage of sub(sup)-script is also very confusing, like in the definition of h_i(u_i). The authors may want to correct typos and improve the presentation. In the conclusion, the authors conclude "We show that the minimax gap shrinks to \eps as the number of generators increases with rate e O(1/\eps)." This is an over-claim, because the authors only proved this under the assumption of concavity of the maximization w.r.t. discriminators.

Finally, the authors may want to provide some simple results of the Stackelberg GAN from the perspective of density approximation, even assuming infinite capacity of the discriminator set, as other GANs does. Whether the distance defined by the maximization problem a distance or divergence. If we exactly minimizing that objective function, do we get the target distribution?

---

> ### Author Response · Authors · 2018-11-09
> **Our theory works broadly including some forms of previous multi-generators GANs as special cases.**
>
> We thank the reviewer for the valuable comments. We will address all concerns from the reviewer in the following form of Q/A.
>
> Q1. (The first paragraph of review) From the algorithm part, I think the algorithm is very similar (and even simpler) than MAD-GAN and MGAN. Therefore, on the algorithm part, the author may want to address the difference between Stackelberg GAN and MAD-GAN and MGAN. On the experiment part, we need to see more comparison between these three methods. In the current experiment, MGAN result is very similar to the proposed method, and MAD-GAN result is missing.
> A1. On one hand, the population form of Stackelberg GAN includes some forms of previous multi-generators GANs as special cases. We believe this is a *plus* of our paper, because it implies that our theory works for broader GAN models, providing a unified and improved framework for multi-generator GANs. Note that in our theory, we make no assumption on the capacity and architecture of discriminator. Thus, our theory even works for more complicated discriminator such as that of MGAN and MAD-GAN, whose theory on equilibrium is missing in their original papers. On the other hand, the empirical losses of Stackelberg GAN and prior GANs are different. Our choice of sampling scheme is flexible as we claimed in the previous post. Furthermore, MGAN requires shared network parameters among various generators, while Stackelberg GAN enjoys free parameters for each generator. To make the paper clearer, we restate the difference among various models in Page 2 of our revised version (the first bullet).
> On the experiment part, we supplement new experiments on MAD-GAN on CIFAR-10 as the reviewer suggested. We did not find existing Inception Score of MAD-GAN on CIFAR-10, so we run it by ourselves. Here is a thorough comparison among MGAN, MAD-GAN, and Stackelberg GAN with the same network capacity and 10 generators. A potential reason of the unsatisfactory performance of MAD-GAN is that the method involves a multi-class discriminator with classes as many as I+1, which leads to an imbalance between real and generated data and the unstable training.
> -----------------------------------------------------------------------------------------
> Model                          Inception Score            Frechet Inception Distance
> ----------------------------------------------------------------------------------------
> MAD-GAN                    6.67+-0.07                        34.10
> MGAN                           7.52+-0.1                          31.34
> Stackelberg GAN        7.62+-0.07                        26.76
>
> Q2. (The second paragraph of review) There are many typos in the paper (and appendix). In the conclusion, the sentence "we show that the minimax gap shrinks to eps as the number of generators increases with rate O(1/eps)" is an over-claim, because the authors only proved this under the assumption of concavity of the maximization w.r.t. discriminators.
> A2. We have tried our best to fix all the typos that we find in the paper and appendix. In particular, we avoid the use of “concave closed hull of a set” by redefining \hat{cl}f:=-\br{cl}(-f). We clarify our use of sub-script in h_i(u_i) by saying “The subscript i in h_i indicates that the function h_i is derived from the i-th generator. The argument of h_i should depend on i, so we denote it by u_i. Intuitively, h_i serves as an approximate convexification of -\phi(\gamma_i,\cdot) w.r.t the second argument due to the conjugate operation”. We also modified the sentence in the conclusion as “we show that the minimax gap shrinks to eps as the number of generators increases with rate O(1/eps), when the maximization problem w.r.t. the discriminator is concave”.
>
> Q3. (The third paragraph of review) The authors may want to provide some simple results of the Stackelberg GAN from the perspective of density approximation. Whether the distance defined by the maximization problem a distance or divergence. If we exactly minimizing that objective function, do we get the target distribution?
> A3. Thanks for the comment. As the reviewer suggested, in Theorem 9 of the revised version we provide new results of Stackelberg GAN from the perspective of density approximation under the standard assumption of Goodfellow’14. Our result shows that Stackelberg GAN can learn a mixture of distributions. This theorem gives a positive answer to the reviewer’s question about whether minimizing the objective function gets the target distribution. For the question concerning whether the distance defined by the maximization problem a distance or divergence, it depends on the choice of function f. For example, when f is the log function, the distance defined by the maximization problem of Stackelberg GAN (i.e., the \tilde{L} in the proof of Theorem 9) is the Jensen-Shannon divergence between the mixture generative distribution and the real distribution.
>
> We are looking forward to a re-evaluation from the reviewer based on our revision.

---

> > ### Comment · AnonReviewer2 · 2018-12-12
> > **my reply**
> >
> > Thank the authors for their detailed reply.
> >
> > Q1. "On one hand, the population form of Stackelberg GAN includes some forms of previous multi-generators GANs as special cases." They are definitely not your special cases, but your formulation is a simplified version of theirs. "On the other hand, the empirical losses of Stackelberg GAN and prior GANs are different." Through our discussion, I got more clear about your formulation. The population loss is the same, but the empirical losses are different. This makes the proof of Theorem 9 the same as before, but minor changes in the training algorithm. This reinforces my impression that the contribution on the algorithmic part is minor.
> >
> > Q2. I do not have time to get back to check whether all typos are corrected and whether the new proof makes sense. I hope that I can read a roughly correct proof (at least typos do not prevent me finishing reading) in my first read.
> >
> > Q3. The authors addressed my Q3 by Theorem 9. From my perspective, Theorem 9 is correct, but the authors' proof is floppy. The equality between (10) and (11) deserves much more words.
> >
> > Overall, I still think that the the contribution on the algorithmic part is minor.

---

### Author Response · Authors · 2018-11-17
**Summary of the revision**

The authors would like to thank the reviewers for bringing up valuable questions/suggestions to improve our paper. We have updated manuscripts to address all concerns from the reviewers appropriately. We summarize our revision below.
----------------------------------------------------------------------------------------------------------------------------------------
Positions                     For which comment                          Revised version
----------------------------------------------------------------------------------------------------------------------------------------
Section D                    Density approximation                     New results for density approximation

Table 1                        Comparison with MAD-GAN            New experiments for MAD-GAN

Sections 2                   Concave closed hull?                        Avoid using the term by redefining \hat{cl} f

Bullet 1, Page 2          Difference with prior GANs            Clarify algorithmic difference

Whole paper              Typos                                                   Fix all typos
-----------------------------------------------------------------------------------------------------------------------------------------

Again, we would like to emphasize that our contributions focus more on the approximate Nash equilibrium in Theorems 1, 3, and Corollary 2, improving over previous best-known bounds. We argue that the algorithmic similarity is an advantage of this paper, which means that our theoretical results work for broader class of multi-generator GANs.

Finally, we would like to mention that the technical contents in the revised version are the same with those in the previous version. We would like to kindly ask reviewers to re-evaluate the paper focusing more on the technical contributions of the paper. Thank you.

---

### Meta-Review · Area_Chair1 · 2018-12-12
**ICLR 2019 decision**

**Confidence:** 4
**Recommendation:** Reject

**Metareview:**

This paper proposes new GAN training method with multi generator architecture inspired by Stackelberg competition in game theory. The paper has theoretical results showing that minmax gap scales to \eps for number of generators O(1/\eps), improving over previous bounds. Paper also has some experimental results on Fashion Mnist and CIFAR10 datasets.

Reviewers find the theoretical results of the paper interesting.  However, reviewers have multiple concerns about comparison with other multi generator architectures, optimization dynamics of the new objective and clarity of writing of the original submission. While authors have addressed some of these concerns in their response reviewers still remain skeptical of the contributions. Perhaps more experiments on imagenet quality datasets with detailed comparison can help make the contributions of the paper clearer.